# MST2 kinase suppresses rDNA transcription in response to DNA damage by phosphorylating nucleolar histone H2B

Dafni Eleftheria Pefani[1,*] (iD), Maria Laura Tognoli[1], Deniz Pirincci Ercan[1,2,†], Vassilis Gorgoulis[3,4,5] & Eric O'Neill[1,6,**] (iD)

## Abstract

The heavily transcribed rDNA repeats that give rise to the ribosomal RNA are clustered in a unique chromatin structure, the nucleolus. Due to its highly repetitive nature and transcriptional activity, the nucleolus is considered a hotspot of genomic instability. Breaks in rDNA induce a transient transcriptional shut down to conserve energy and promote rDNA repair; however, how nucleolar chromatin is modified and impacts on rDNA repair is unknown. Here, we uncover that phosphorylation of serine 14 on histone H2B marks transcriptionally inactive nucleolar chromatin in response to DNA damage. We identified that the MST2 kinase localises at the nucleoli and targets phosphorylation of H2BS14p in an ATM-dependent manner. We show that establishment of H2BS14p is necessary for damage-induced rDNA transcriptional shut down and maintenance of genomic integrity. Ablation of MST2 kinase, or upstream activators, results in defective establishment of nucleolar H2BS14p, perturbed DNA damage repair, sensitisation to rDNA damage and increased cell lethality. We highlight the impact of chromatin regulation in the rDNA damage response and targeting of the nucleolus as an emerging cancer therapeutic approach.

**Keywords** ATM; chromatin; DNA damage; RASSF1A; rDNA transcription
**Subject Categories** Chromatin, Epigenetics, Genomics & Functional Genomics; DNA Replication, Repair & Recombination; Post-translational Modifications, Proteolysis & Proteomics
**The EMBO Journal (2018) 37: e98760**

## Introduction

Cells are continuously exposed to endogenous and environmental conditions (e.g. cellular respiration or ionising radiation) that promote breaks or lesions in DNA which can lead to genomic instability. Efficient recognition of DNA damage and lesion repair is orchestrated by the DNA damage response. As DNA is organised to chromatin, dynamic changes of histone modifications are critical for regulating double-strand break (DSB) repair (Kumar *et al*, 2012). Recent studies have shown that the position of a DNA break relative to chromatin determines the choice of repair pathway and therefore influences the impact of the break on genomic stability (Lemaitre *et al*, 2014; Harding *et al*, 2015; Ryu *et al*, 2015; van Sluis & McStay, 2015). The genetic loci encompassing the ribosomal genes (rDNA) are the largest repetitive elements of the human genome and are organised within the nucleolus for direct coupling to ribosome biogenesis. The recombinogenic nature of the rDNA repeats, together with high levels of ribosomal gene transcription, results in the nucleolus being a hotspot of genomic instability (Gaillard & Aguilera, 2016; Warmerdam *et al*, 2016). Concomitantly, translocations involving the rDNA repeats are amongst the most common events observed in cancers (Stults *et al*, 2009). Therefore, understanding how DNA damage responses are conducted in this nuclear subdomain is vital to interpret the contribution of genomic instability to cancer. In response to nuclear DNA damage response (DDR) activation or localised damage within the nucleolus a transient polymerase I (Pol I), ATM kinase-dependent transcriptional shut down takes place (Kruhlak *et al*, 2007; Larsen *et al*, 2014). ATM activity results in Pol I displacement and inhibition of the kinase abrogates the Pol I transcriptional shut down (Kruhlak *et al*, 2007). This transcriptional inhibition saves energy for repair and protects from collision of transcription and repair machineries within this highly transcribed locus. Observations in yeast reveal that high rRNA

1   CRUK/MRC Institute for Radiation Oncology, Department of Oncology, University of Oxford, Oxford, UK
2   Radboud University, Nijmegen, The Netherlands
3   Laboratory of Histology and Embryology, Medical School, National and Kapodistrian University of Athens, Athens, Greece
4   Biomedical Research Foundation of the Academy of Athens, Athens, Greece
5   Faculty of Biology, Medicine and Health, Manchester Academic Health Centre, University of Manchester, Manchester, UK
6   Systems Biology Ireland, University College Dublin, Dublin 4, Ireland
    *Corresponding author. Tel: +44 1865 617360; E-mail: dafni.pefani@oncology.ox.ac.uk
    **Corresponding author. Tel: +44 1865 617321; E-mail: eric.oneill@oncology.ox.ac.uk
    †Present address: The Francis Crick Institute, Chromosome Segregation Laboratory, London, UK

transcription rates are associated with DNA repair defects and genome instability (Ide *et al*, 2010), indicating that DNA damage-induced Pol I transcriptional shut down is important to maintain overall genome integrity. Recent studies have also shed light on how DSBs within nucleolar chromatin are processed and how rDNA repair impacts on rDNA transcription (Harding *et al*, 2015; Warmerdam *et al*, 2016). However, how nucleolar chromatin is organised under these conditions remains poorly understood.

Herein, we show that in response to DNA damage, there is increased phosphorylation of histone H2B at serine 14 (H2BS14p). H2BS14p has been shown to lead to chromatin condensation both *in vitro* and *in vivo* and has been described as a feature of apoptotic chromatin (de la Barre *et al*, 2001; Cheung *et al*, 2003). The histone mark has also been identified in ionising radiation-induced DNA damage foci co-localising with the major double-strand break marker γH2Ax (Fernandez-Capetillo *et al*, 2004). Previous studies have shown that apoptotic H2BS14p is established by the MST1 Ser/Thr kinase (Cheung *et al*, 2003; Ahn *et al*, 2005). Here, we show that MST1 is dispensable for H2BS14p nucleolar accumulation in response to DNA damage, in contrast, the MST2 paralogue is localised in the nucleolus and specifically targets nucleolar H2BS14p. We show that H2BS14p establishment is an integral part of the ATM nucleolar signalling and that the RASSF1A scaffold, a previously characterised ATM target and activator of the MST2 kinase (Hamilton *et al*, 2009), is necessary for the response. We show that in the absence of the ATM-RASSF1A-MST2 axis, the lack of H2BS14p establishment results in perturbed transcriptional silencing of nucleolar chromatin in the presence of rDNA damage. Most importantly, lack of H2BS14p leads to persistent nucleolar damage and decreased viability linking chromatin modifications with Pol I transcriptional shut down and providing a new mechanistic insight on how cells respond to nucleolar double-strand breaks.

# Results

## Phospho-histone H2B marks nucleolar chromatin in response to ionising radiation-induced DNA damage

Phosphorylation of histone H2B at serine 14 (H2BS14p) is a well-established apoptotic mark. *In vivo* and *in vitro* studies show that H2BS14p promotes chromatin condensation, a major feature of apoptotic cells. In agreement with previous reports, we were able to detect H2BS14p in apoptotic cells (Cheung *et al*, 2003; Ahn *et al*, 2005) and at the midbody of cells in telophase (Rinaldo *et al*, 2012), while the mark was absent from mitotic chromatin (Cheung *et al*, 2003; Fig EV1A–C). Interestingly, after exposure of HeLa cells to ionising radiation (γIR), we noticed accumulation of H2BS14p in the DAPI poor staining regions that mark the nucleoli (Fig 1A). The accumulation of H2BS14p in the nucleoli is rapid and transient, as 1 h after exposure to DNA damage the signal was no longer detectable (Fig 1A and B). We further verified the nucleolar localisation of H2BS14p by co-staining with nucleolin, the major nucleolar protein of growing eukaryotic cells (Figs 1C and EV1D and E) using two different antibodies whose specificity was tested against a non-phosphorylatable H2B variant, H2BS14A-GFP (Fig EV1B). Also, a transient increase in the H2BS14p was observed in total cell extracts

from cells exposed to γIR (Fig 1D). In addition to HeLa cells, DNA damage-induced nucleolar H2BS14p was detectable in U2OS cells (Fig EV1F) as well as in primary human bronchial epithelial cells (HBECS; Fig EV1G), suggesting that the accumulation of the mark is a ubiquitous nucleolar response to DNA damage. To exclude the possibility that the increase in the H2BS14p is due to transient accumulation of histone H2B in the nucleolus, we examined total H2B levels and did not observe any significant differences between irradiated and control cells (Fig 1E). These data suggest that H2BS14p marks nucleolar chromatin after exposure to γIR and induction of DNA damage. Therefore, we identify H2BS14p as a feature of nucleolar chromatin in response to DNA damage, in line with previous studies showing that histone modifications within nucleolar chromatin might be regulated differently than other areas of the genome (Tessarz *et al*, 2014).

## MST2 Ser/Thr kinase phosphorylates nucleolar H2B on serine 14 in response to ionising radiation

The Mammalian Sterile20 like kinase MST1 has been described to phosphorylate histone H2BS14p in response to apoptotic stimuli both *in vitro* and *in vivo* (Cheung *et al*, 2003; Ahn *et al*, 2005; Bitra *et al*, 2017). This appears developmentally conserved as the Ste20 orthologue similarly mediates the phosphorylation of histone H2B at the equivalent Ser10 residue in yeast (Cheung *et al*, 2003; Ahn *et al*, 2005). We were not able to detect any significant accumulation of MST1 kinase within the nucleolus, either in normal cycling cells or after exposure to γIR (Fig 2A). However, we did observe a significant nucleolar fraction of the closely related MST2 kinase (Fig 2B), which was specific as the signal was absent in MST2 RNAi-treated cells (Fig EV2A). MST2 nucleolar localisation did not appear to vary during cell cycle as it was detected in both positive and negative EdU incorporating cells (Fig 2C), nor did we observe any significant changes upon exposure to γIR (Fig 2D and E). However, we did detect an increased interaction between MST2 and histone H2B after induction of DNA damage (Fig 3A). In order to test whether MST2 binds to free H2B pools or interacts with chromatin-bound nucleosomes, we looked for association of histones H2A, H3 and H4 in MST2 immunoprecipitates and found increased presence of all core histones upon exposure to γIR (Fig EV2B). To determine whether there was enrichment of MST2 within the rDNA repeats in the nucleolus, we used chromatin immunoprecipitation (ChIP) followed by quantitative real-time PCR (qPCR) and looked for MST2 occupancy at the promoter region upstream from the start site (H0), at the coding region (H1) and the intragenic spacer (H18) of the rDNA repeat. As a negative control we used a primer set for GAPDH (Fig 3B). MST2 displayed increased association across the rDNA repeats after treatment with γIR, suggesting a DNA damage-induced recruitment of MST2 into nucleolar chromatin. Importantly, treating cells with siRNA against MST2 resulted in a failure to establish H2BS14p in response to DNA damage, whereas ablation of MST1 had no significant effect (Figs 3C and D, and EV2C). Lack of MST2 expression results in a reduction of the H2BS14p signal rather than a kinetic delay as up to 4 h post-exposure to γIR we were unable to detect H2BS14p in siRNA-treated cells (Fig EV2D). The above data indicate that nucleolar H2B is a target of the MST2 and not the MST1 kinase, potentially due to limited availability of MST1 in the nucleolus.

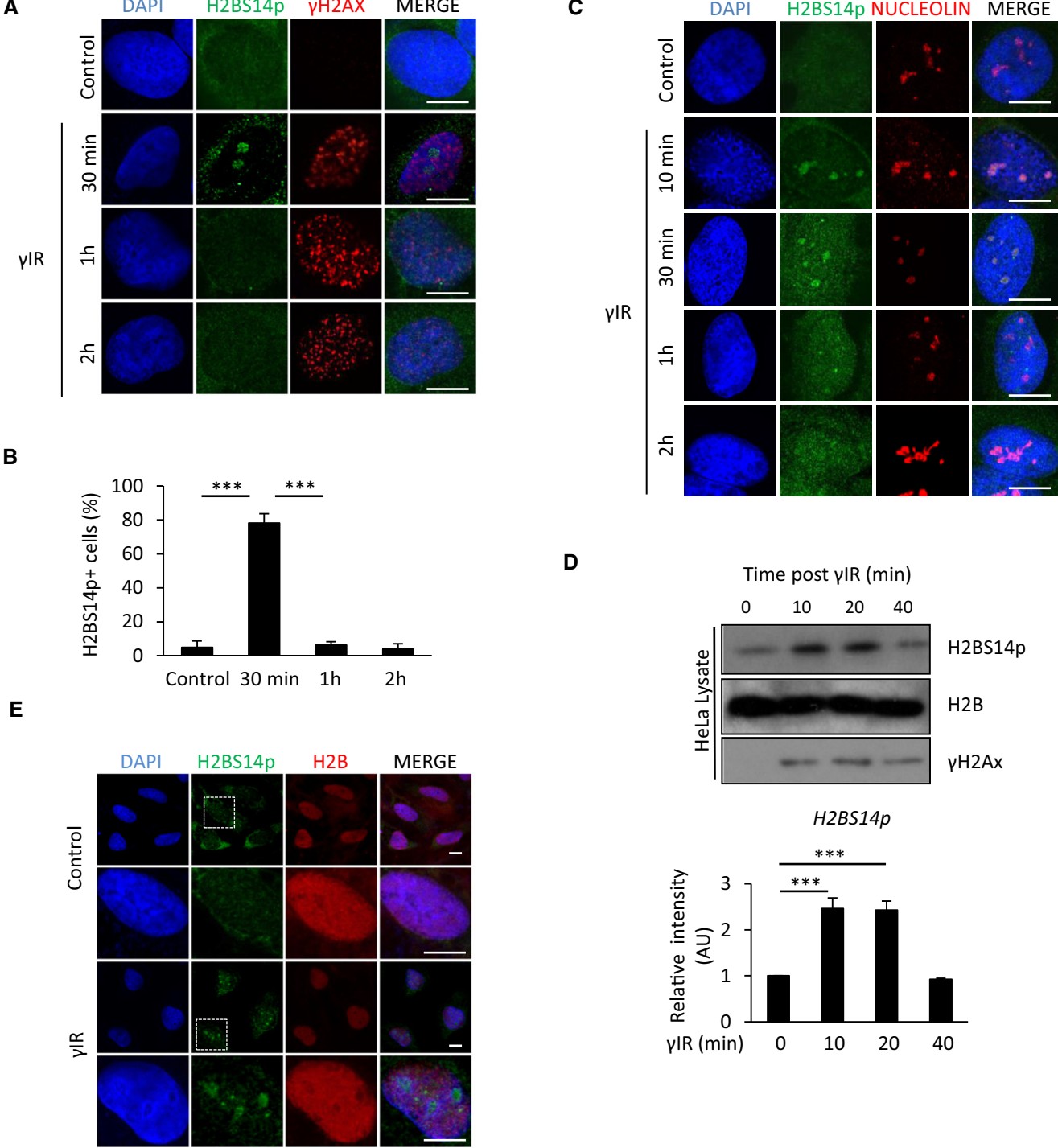

**Figure 1. Nucleolar histone H2B gets phosphorylated at serine 14 in response to γIR.**

A   HeLa cells were exposed to 5 Gy γIR, fixed at indicated time points and stained for γH2AX and H2BS14p.

B   Quantification of HeLa cells with nucleolar staining for H2BS14p at the indicated times after exposure to γIR. Error bars represent SD and derive from three independent experiments.

C   HeLa cells exposed to γIR, fixed at the indicated times and stained for H2BS14p and nucleolin.

D   Western blot analysis and quantification for H2BS14p at the indicated times after exposure to γIR. Error bars represent standard deviation and derive from two independent experiments.

E   HeLa cells were treated with γIR and fixed 10 min after exposure. Higher magnification images of boxed areas are shown.

Data information: DNA was stained with DAPI. Scale bars 10 μm. Two-tailed Student's *t*-test was used for statistical analysis. ***$P < 0.001$.

Source data are available online for this figure.

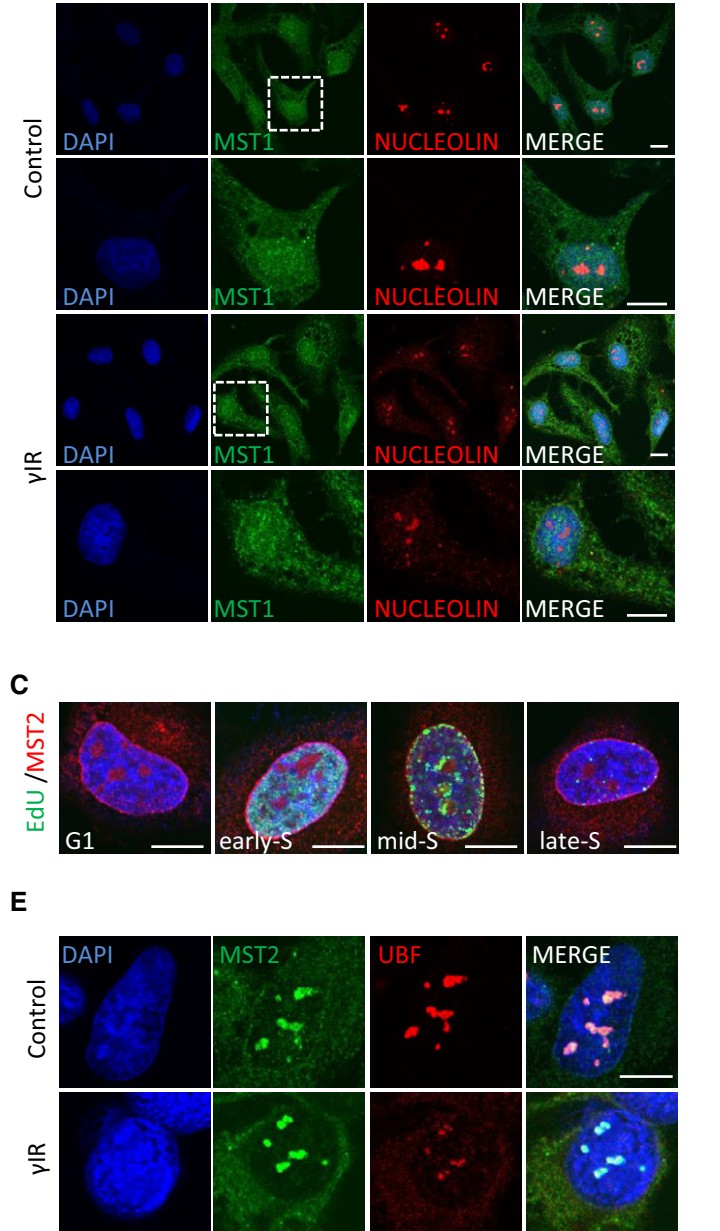

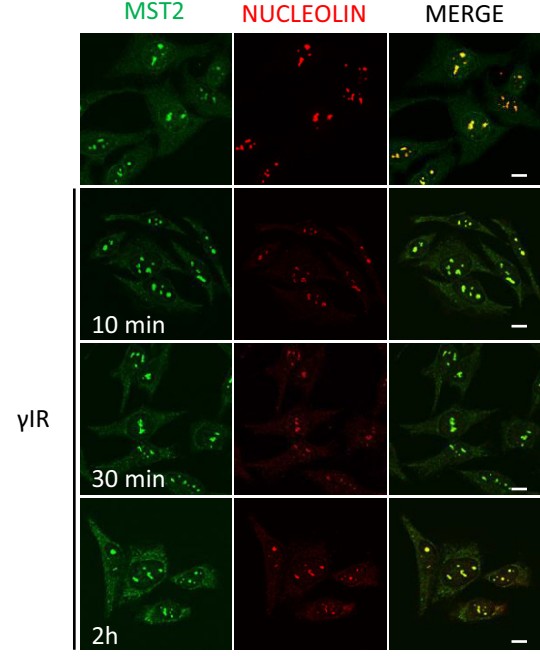

**Figure 2. MST2 localises in the nucleoli independent of the presence of damage.**

A HeLa cells were exposed to γIR, fixed 10 min after and stained for MST1 localisation. Higher magnification images of boxed areas are shown.

B HeLa cells were fixed and stained with the indicated antibodies.

C HeLa cells were treated with EdU for 20 min, fixed and stained for MST2. EdU containing DNA was visualised using the Click-IT EdU imaging kit (Invitrogen).

D HeLa cells were exposed to γIR, fixed at the indicated time points and stained for MST2 and nucleolin.

E HeLa cells were exposed to γIR, fixed after 10 mins and stained for MST2 and UBF.

Data information: DNA was stained with DAPI. Scale bars at 10 μm.

## H2BS14p promotes rDNA transcriptional shut down in response to ionising radiation

ATM signalling has been shown to suppress transcription at the sites of damage (Kruhlak *et al*, 2007; Harding *et al*, 2015). In response to rDNA damage, there is an ATM-dependent transcriptional inhibition of polymerase I (Pol I) activity (Kruhlak *et al*, 2007; Larsen *et al*, 2014). The rDNA transcriptional shut down is maximal within 10 min of exposure to γIR, and transcription is restored approximately 1 h after initial genotoxic stress (Kruhlak *et al*, 2007; Larsen *et al*, 2014). We noticed that the kinetics for transient Pol I shut down were very similar to the rate of nucleolar

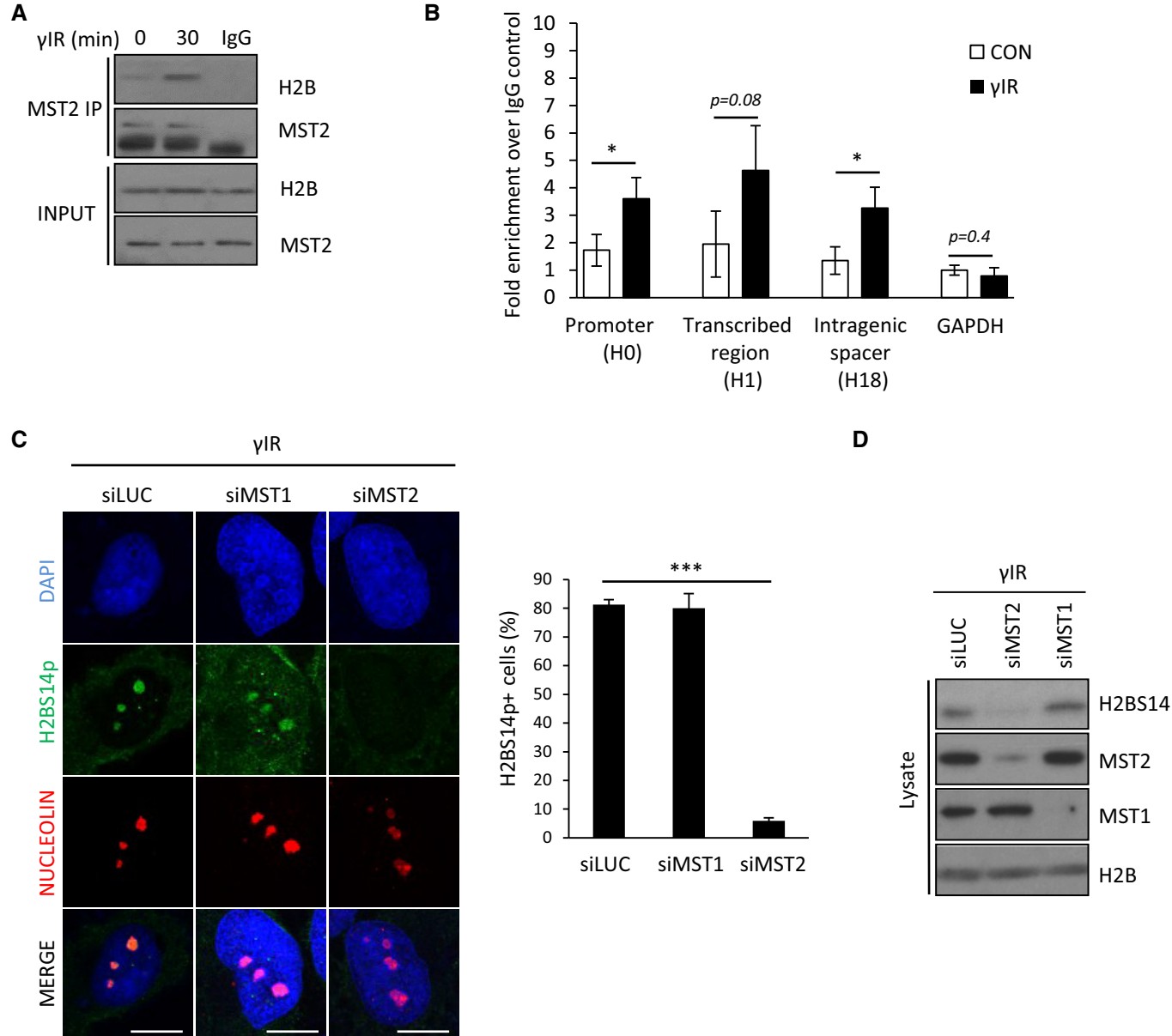

**Figure 3. MST2 phosphorylates nucleolar H2B at serine 14 in response to γIR.**

A   HeLa cells were exposed to γIR and lysed, and anti-MST2 IgG or control antibody was used to immunoprecipitate proteins from total cell extracts prior to Western blotting with indicated antibodies.
B   U2OS cells were exposed or not to γIR and subjected to ChIP–qPCR to analyse the enrichment of MST2 over IgG control. Primer sets were targeted at the promoter (H0), coding region (H1) or the intragenic spacer between rDNA repeats (H18). Error bars derive from three independent experiments and represent the SEM.
C   HeLa cells were treated with the indicated siRNAs, exposed to γIR, fixed and stained with the indicated antibodies. Representative images and quantification of H2BS14p-positive cells in each condition are shown. Error bars represent the SD and derive from three independent experiments. DNA was stained with DAPI. Scale bars at 10 μm.
D   HeLa cells were treated with the indicated siRNAs, exposed to γIR and lysates prepared prior to Western blotting for the indicated antibodies.

Data information: Two-tailed Student's *t*-test was used for statistical analysis. *$P < 0.05$, ***$P < 0.001$.
Source data are available online for this figure.

H2BS14p establishment (Fig 1B). We similarly detected reduction of Pol I transcription in response to γIR, assessed by pre-rRNA transcript abudance and 5-EU RNA labelling in an ATM-dependent manner (Figs 4A and EV2E). To test whether MST2 regulates rDNA transcription under these conditions, we depleted MST2 and

measured Pol I activity using the same readouts. We observed that γIR-mediated suppression of pre-rRNA transcripts and 5-EU incorporation was affected in cells that lack MST2 (Figs 4B and C, and EV2F). This appears specific to MST2 as depletion of MST1 did not result in significant changes under these conditions (Fig 4B, single

cell in Fig 4C and population in Fig EV2F). We further verified these results using a second siRNA oligo against MST2 (Fig EV2G) that also resulted in lack of H2BS14p establishment (Fig EV2H) and increased Pol I transcription (Fig EV2I and J).

To directly link failure to establish H2BS14p with impaired nucleolar transcription in the absence of MST2 kinase, we asked whether expression of a phospho-mimetic H2BS14D derivative, in which the S14 residue was replaced with an aspartic acid to mimic constitutive phosphorylation, would result in decreased Pol I transcription (Fig EV3A). Expression of H2BS14D-GFP resulted in decreased pre-rRNA transcripts in both HeLa and U2OS cells (Figs 4D and EV3B and C). In contrast, cells transfected with the non-phosphorylatable variant, H2BS14A-GFP (Fig EV3A), show higher levels or pre-rRNA transcripts after exposure to γIR (Figs 4E and EV3D) compared with control cells in agreement with H2BS14p promoting decreased rDNA transcription in response to DNA damage. To determine whether establishment of the H2BS14p is a causative for rDNA transcriptional shut down and not simply a consequence of Pol I inhibition, we used a Pol I inhibitor (CX-5461) and checked for nucleolar H2BS14p in undamaged cells. Phosphorylation of H2B was undetectable in the presence of the inhibitor (Fig EV3E), confirming that DNA damage-induced MST2-dependent establishment of H2BS14p is a mark of nucleolar chromatin that occurs upstream of transcriptional silencing, rather than induced by Pol I inhibition.

It has been reported that the establishment of H2BS14p in the apoptotic chromatin is recognised by the regulator of chromosome condensation (RCC1; Wong *et al*, 2009). RCC1 was originally described to regulate the onset of chromosome condensation (Ohtsubo *et al*, 1989). To test whether nucleolar H2BS14p would result in stabilisation of RCC1 on nucleolar chromatin, we checked for RCC1 nucleolar recruitment soon after exposure to γIR (Fig 4F). In untreated cells, we could not observe co-localisation of RCC1 with nucleolin. However, 10 min after exposure to γIR, we could see accumulation of RCC1 in the nucleolus. In agreement with an H2BS14p-dependent recruitment, we observed loss of the RCC1 nucleolar signal 1 h after induction of γIR (Figs 1B and C, and 4F).

The above data suggest that MST2-dependent establishment of nucleolar H2BS14p in response to DNA damage regulates rDNA transcription promoting chromatin compaction via recruitment of RCC1.

## Nucleolar H2BS14p depends on ATM signalling

To gain further mechanistic insight on the DNA damage-induced phosphorylation of H2BS14 in the nucleolus, we next addressed the activation signal for the MST2 kinase. MST2 activity is increased in response to genotoxic stress via ATM- or ATR-mediated phosphorylation of serine 131 on the adaptor protein RASSF1A. This promotes RASSF1A homodimerisation which increases the local concentration of MST2 and allows transphosphorylation of kinase activation loop residues required for substrate activity (Hamilton *et al*, 2009; Pefani *et al*, 2014). RASSF1A interacts with MST2 through SARAH domain interactions, and recent studies have shown that the RASSF1 SARAH domain increases MST kinase activity against H2B *in vitro* (Bitra *et al*, 2017). ATM has a major role in the DNA damage imposed transcriptional shut down in the nucleolus including directly regulating Pol I (Kruhlak *et al*, 2007; Larsen *et al*, 2014). To assess whether ATM also regulates the nucleolar chromatin organisation under these conditions, we used a specific ATM kinase inhibitor (KU55933) and looked for nucleolar H2BS14p establishment. In contrast to control cells, we were not able to detect nucleolar H2BS14p in HeLa cells that were treated with the ATM inhibitor prior to exposure to γIR (Fig 5A). MST2 activity depends on autophosphorylation of a unique threonine residue Th180 (Ni *et al*, 2013). Therefore, we checked for MST2 autoactivation upon exposure to γIR in the presence or absence of ATM inhibition (Fig 5B). As previously shown (Hamilton *et al*, 2009), we observed increased MST2 autophosphorylation in response to γIR in an ATM-dependent manner (Fig 5B). In agreement with ATM acting upstream of MST2 and regulating rDNA transcription via activating several responses (Ciccia *et al*, 2014; Larsen *et al*, 2014), we observed a more profound impact on rDNA transcription in the absence of ATM compared with MST2 deletion alone and combination of both did not have a greater impact on rDNA silencing (Fig 5C). Recent studies have shown involvement of DNA-PK and PARP in Pol I and Pol II transcriptional repression in the presence of DNA damage (Pankotai *et al*, 2012; Calkins *et al*, 2013; Awwad *et al*, 2017). We therefore checked whether inhibition of DNA-PK or PARP could affect MST2 kinase activity but did not observe any impact (Fig EV3F). Therefore, we concluded that MST2 activation is part of the ATM-mediated response to achieve Pol I inhibition in response to DNA damage.

---

Figure 4. **MST2 regulates nucleolar transcription in response to γIR via H2BS14 phosphorylation.**

A   Relative pre-rRNA expression in HeLa cells at the indicated times after exposure to γIR. Expression of pre-rRNA was normalised against GAPDH. Error bars represent the SD and derive from two independent experiments.

B   HeLa cells were treated with the indicated siRNAs and exposed to γIR. 20 min after exposure pre-rRNA expression was normalised against GAPDH. Error bars represent the SD and derive from two independent experiments.

C   HeLa cells were treated with the indicated siRNAs, exposed to γIR and treated with 0.5 mM 5-EU for 20 min. 5-EU incorporation was quantified, and representative images from each condition are shown here and in Fig EV2F. Quantification of data derived from three independent experiments is shown on the right. Middle line represents the median and the boxes 25th and 75th percentiles. The whiskers mark the smallest and largest values.

D   HeLa cells were transfected with H2B-GFP or H2BS14D-GFP. pre-rRNA abudance was assessed with qPCR. Expression of pre-rRNA was normalised against GAPDH. Error bars derive from two independent experiments and represent the SD.

E   HeLa cells were transfected with H2B-GFP or H2BS14A-GFP and exposed to γIR. pre-rRNA abudance was assessed with qPCR. Expression of pre-rRNA was normalised against GAPDH. Error bars derive from two independent experiments and represent the SD.

F   HeLa cells were exposed to γIR, fixed at the indicated times and stained for RCC1 and nucleolin. Representative images (left) and quantification (right) of cells with nucleolar staining of RCC1 are shown. Error bars represent the SD and derive from three independent experiments.

Data information: DNA was stained with DAPI. Scale bars at 10 μm. Two-tailed Student's *t*-test (A, B, D–F) or Mann–Whitney test (C) was used for statistical analysis.
*$P < 0.05$, **$P < 0.01$, ***$P < 0.001$.

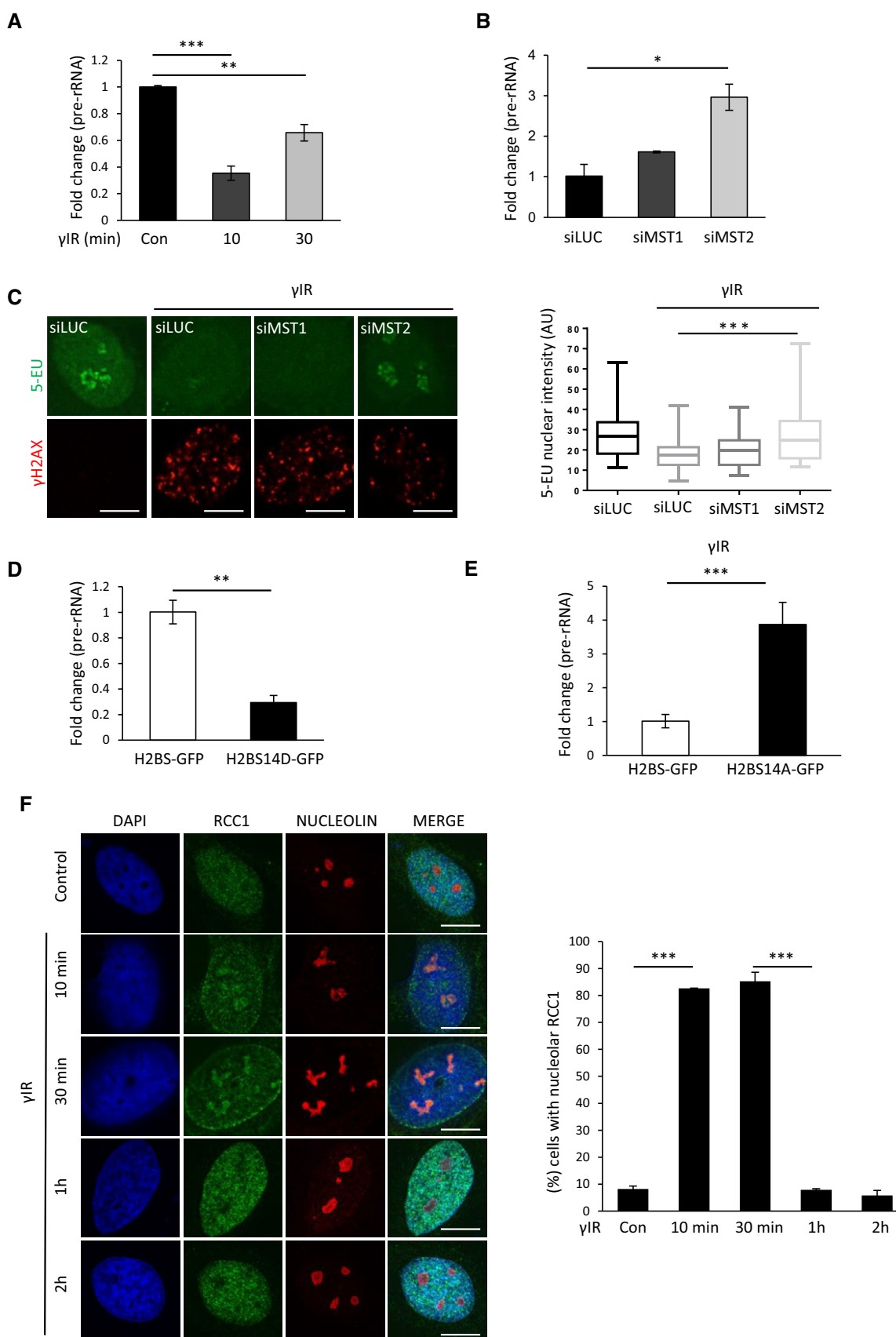

**Figure 4.**

To examine the necessity of adaptor proteins for MST2 DNA damage-induced nucleolar activity, we depleted cells for RASSF1A or Salvador 1 (SAV1), an MST1/2 adaptor known to stimulate MST kinase activity during development (Yin *et al*, 2013). Similarly to MST2 deletion, RASSF1A mRNA knock-down abolished establishment of nucleolar H2BS14p upon exposure to γIR, suggesting that ATM is likely to signal to nucleolar MST2 via RASSF1A. Loss of SAV1 did not affect nucleolar H2BS14p indicating that different adaptor proteins activate MST depending on the cellular context and subcellular localisation (Figs 5D and EV3G). Indeed, cellular fractionation experiments show that a substantial level of RASSF1A and MST2 was found in the nucleolar enriched fraction. In contrast, SAV1 and MST1 were mainly found to be cytoplasmic (Fig 5E).

### The ATM-RASSF1A-MST2 axis promotes nucleolar H2BS14p in the presence of rDNA DSBs

rDNA represents ~0.5% of the genome; therefore, damage within the nucleolus represents a minority of the total damage present in irradiated nuclei. To study in detail the response of nucleoli to DSBs introduced into rDNA, we took advantage of the homing endonuclease from *Physarum polycephalum* (I-PpoI) that recognises a sequence within the 28S-rDNA coding region of each of the approximately 300 rDNA repeats and 13 other sites at the human genome (Muscarella *et al*, 1990). This allowed us to study a response of extensive DSBs that take place mainly in the nucleolus. In line with previous observations, 6 h post-transfection of V5 epitope-tagged I-PpoI mRNA, ~80% cells undergo nucleolar transformation and form γH2Ax/UBF-positive nucleolar caps (Figs 6A and EV4A). As expected, exogenous I-PpoI mRNA expression is no longer detectable 24 h post-transfection and the majority of damage appears repaired at this time, i.e. loss of γH2Ax signal from the nucleolar caps (Fig 6A). While I-PpoI efficiently induces γH2Ax, introduction of a catalytically inactive I-PpoI mutant (H98A) fails to induce rDNA damage and nucleolar reorganisation (Figs 6B and EV4A). In agreement with previous studies, we detect lack of 5-EU incorporation in the nucleolus shortly after exposure to I-PpoI WT but not I-PpoI-H98A (Fig 6B). We also observed that inhibition of ATM kinase completely rescues the transcriptional shut down under these conditions (Harding *et al*, 2015; van Sluis & McStay, 2015) (Fig EV4B). This transcriptional inhibition persists for up to 20 h, after which I-PpoI expression is lost and the majority of rDNA is repaired (Figs 6A and EV4C). We next checked for establishment of

nucleolar H2BS14p under these conditions of targeted damage to rDNA. Nucleolar H2BS14p is found in cells transfected for I-PpoI, but not in cells expressing the catalytically inactive mutant (Fig 6C). In agreement with our γIR data, we also observed nucleolar H2BS14p to be dependent on ATM activity in response to rDNA breaks introduced by I-PpoI (Fig 6D). Correlating with the rDNA transcriptional shut down kinetics upon rDNA DSBs with I-PpoI, we observe that nucleolar H2BS14p is lost 24 h post-mRNA transfection (Figs 6A and EV4D). Replicating the phenotype of irradiated cells, we also observed that cells failed to establish H2BS14p (Fig 6E) or restrict 5-EU incorporation upon I-PpoI transfection after deletion of the MST2 kinase or the adaptor protein RASSF1A (Figs 6F and EV4E and F). Moreover, overexpression of the H2BS14A-GFP variant results in higher rDNA transcription in the presence of rDNA DSBs assessed by qPCR (Figs 6G and EV4G). Previous reports have shown that nucleolar reorganisation in the presence of persistent DSBs introduced by I-Ppo I is linked with lack of Pol I transcription under these conditions (Harding *et al*, 2015; van Sluis & McStay, 2015). Indeed, in the presence of ATM inhibition, where rDNA transcription is reconstituted, we see a complete rescue of nucleolar segregation upon I-Ppo I expression (Fig EV5A). MST2 deletion results in a significant reduction in the fully segregated and increase in partially segregated nucleoli compared with control-treated cells, indicative of the higher rDNA transcription that takes place in the absence of the kinase (Fig EV5A).

An interesting observation is that H2BS14p does not co-localise with γH2Ax at the nucleolar caps, but rather marks H2B at the nucleolar interior (Fig EV5B), suggesting that detected H2BS14p does not localise in the nucleolar caps where rDNA breaks are repaired via homologous recombination (HR; van Sluis & McStay, 2015).

### MST2 promotes survival in the presence of rDNA DSBs

We next established the biological significance of rDNA transcription arrest upon DNA damage. To determine the consequence of failing to restrict rDNA transcription via H2BS14p, we induced I-PpoI cleavage of rDNA and tested cell survival by ability to form colonies. Previous studies have shown that these breaks can be highly toxic in RPE-1 cells resulting in low survival (Warmerdam *et al*, 2016). We also observed high lethality of RPE-1 cells exposed to rDNA damage, but interestingly, HeLa cells were more resistant to these toxic lesions (Fig EV5C). To assess the impact of H2BS14p,

---

**Figure 5. Nucleolar H2BS14p establishment depends on the ATM-RASSF1A-MST2 axis.**

A HeLa cells were treated with DMSO or 10 μM of the ATM inhibitor KU55933 and exposed to γIR. After 10 min, cells were fixed and stained with the indicated antibodies. Representative images and quantification (right) of H2BS14p-positive cells are shown. Error bars represent SD and derive from three independent experiments.

B HeLa and U2OS cells were treated with DMSO or exposed to γIR in the presence or absence of KU55933, 10 μM. Cell lysates were isolated, analysed by Western blotting and probed for the indicated antibodies.

C HeLa cells were treated or not with siRNA against MST2 and/or KU55933 and exposed to γIR. 20 min post-γIR pre-rRNA expression was assessed with qPCR. Data derive from two independent experiments and represent the SD.

D HeLa cells were treated with the indicated siRNAs and exposed to γIR. 10 min after, cells were fixed and stained for the indicated antibodies. Representative images (one cell here, larger field of view in Fig EV3G) and quantification (below) of H2BS14p-positive cells are shown. Error bars represent SD and derive from two independent experiments.

E Western blot analysis of HeLa cell fractionation with indicated antibodies.

Data information: DNA was stained with DAPI. Scale bars at 10 μm. Two-tailed Student's *t*-test was used for statistical analysis. **$P < 0.01$, ***$P < 0.001$.
Source data are available online for this figure.

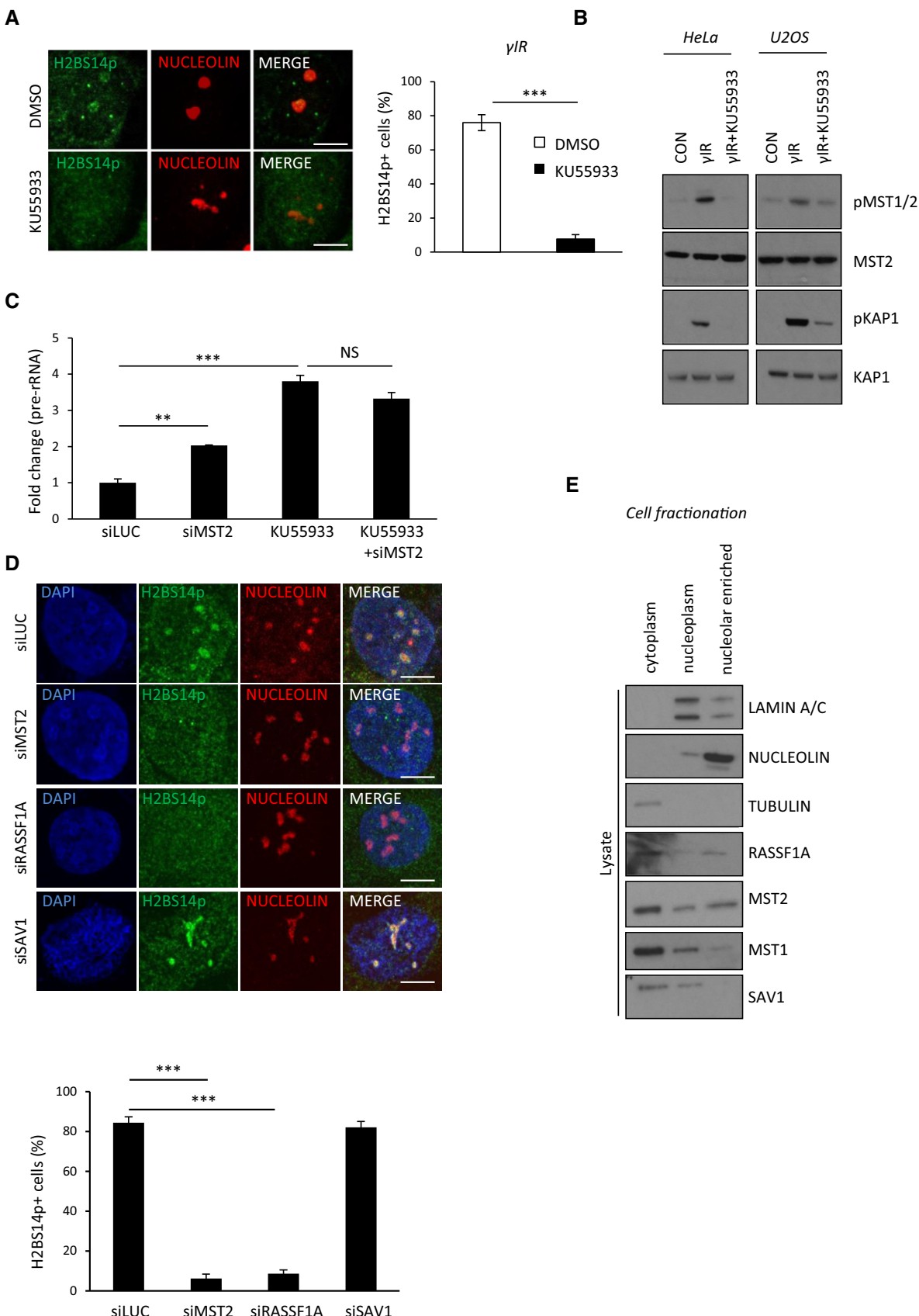

**Figure 5.**

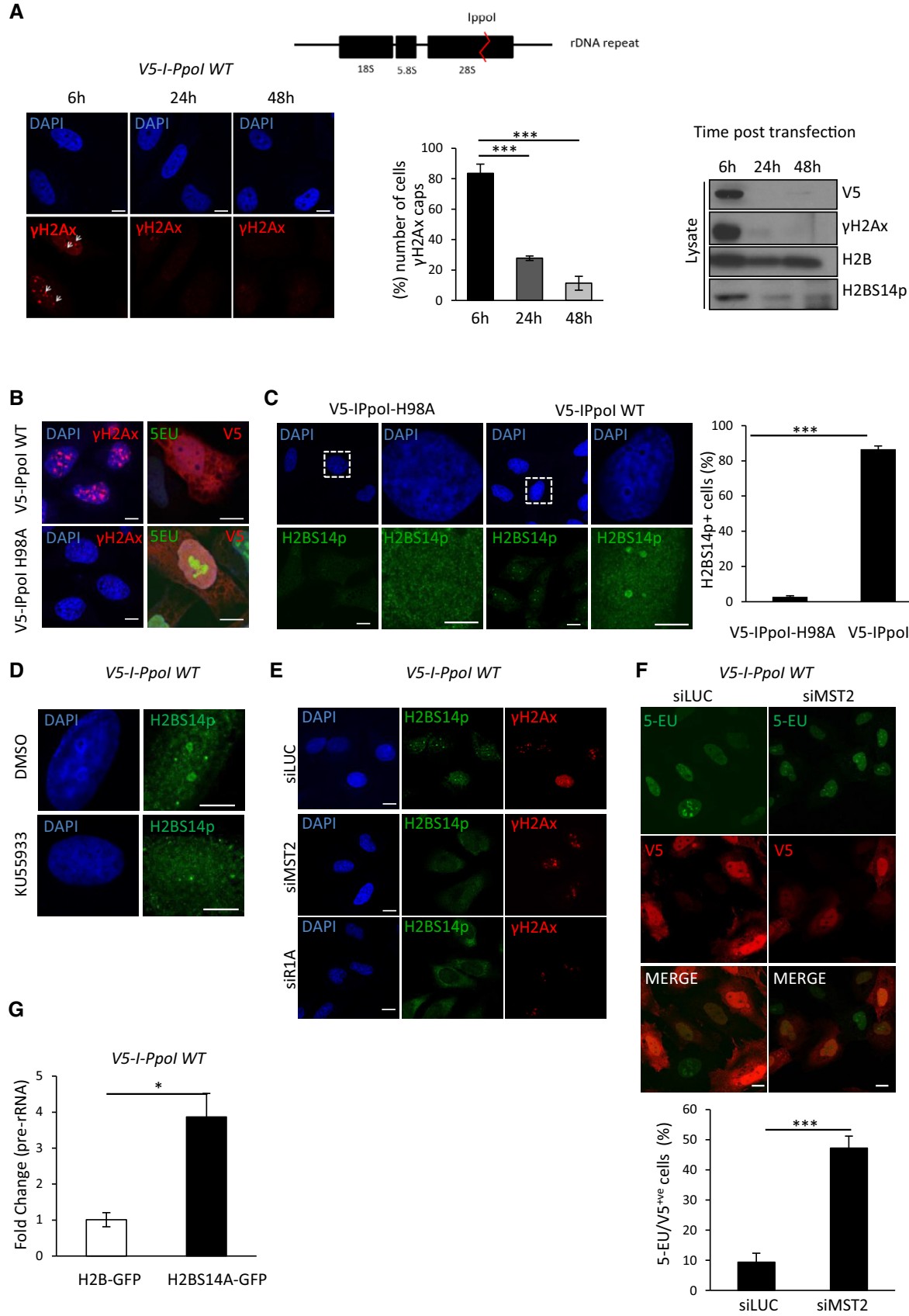

**Figure 6.**

**Figure 6.  rDNA DSBs result in MST2-dependent transcriptional shut down.**

A   I-PpoI recognises a 15 bp sequence in the rDNA repeat (top); *In vitro* transcribed mRNA of a V5 epitope-tagged derivative was directly transfected into HeLa cells. Cells were lysed at the indicated times and analysed with Western blot for the indicated antibodies (lower right). rDNA repair was measured by the presence of γH2AX. Representative images (lower left) and quantification (lower middle) of cells with γH2AX-positive nucleolar caps are shown. Arrowheads point at γH2AX-positive nucleolar caps. Error bars represent SD and derive from three independent experiments.

B   HeLa cells were transfected with mRNA from V5-I-PpoI WT or catalytically inactive I-PpoI H98A. 6 h post-mRNA transfection accumulation of γH2AX and 5-EU incorporation was assessed.

C   I-PpoI WT or I-PpoI H98A mRNA was transfected in HeLa cells, 6 h post-transfection cells was fixed and stained for H2BS14p. Boxed areas are shown in higher magnification. Representative images and quantification of H2BS14p-positive cells are shown. Error bars represent the SD and derive from three independent experiments.

D   HeLa cells were treated or not with 10 μM ATM inhibitor (KU55933), transfected with I-PpoI WT mRNA as above, followed by fixation and staining for H2BS14p.

E   HeLa cells were initially transfected with the indicated siRNAs and I-PpoI WT mRNA introduced after 48 h, cells were stained for the indicated antibodies.

F   HeLa cells were initially transfected with siMST2 or control siRNA and I-PpoI WT mRNA introduced after 48 h. Six hours post-mRNA transfection cells were assessed for I-PpoI expression and 5-EU incorporation. Quantifications and representative images are shown. Error bars represent the SD and derive from three independent experiments.

G   HeLa cells were transfected with H2B-GFP or H2BS14A-GFP. rDNA DSBs were introduced transfecting by I-PpoI-WT mRNA. Pre-rRNA expression relative to GAPDH was assessed with qPCR. Error bars represent the SD and derive from two independent experiments.

Data information: Scale bars 10 μm. Two-tailed Student's *t*-test was used for statistical analysis. *$P < 0.05$, ***$P < 0.001$.
Source data are available online for this figure.

we depleted MST2 or RASSF1A and checked for cell viability in cells transfected with I-PpoI versus the catalytically inactive mutant. Loss of RASSF1A or MST2 significantly reduced the ability of RPE-1 and HeLa cells to survive nucleolar rDNA damage, whereas restriction of MST1 expression did not (Fig 7A). To understand whether decreased cell survival in the absence of H2BS14p was due to impaired rDNA repair, we monitored γH2Ax levels at the nucleolar caps. Depletion of MST2 prior to rDNA damage resulted in γH2Ax maintenance up to 48 hours after induction of damage, indicating failure to resolve rDNA DSBs in the absence of H2BS14p (Fig 7B). Taken together, the above data highlight H2BS14p as a specific nucleolar chromatin modification in response to DNA damage in the rDNA repeats. Our model proposes that H2BS14p supports rDNA transcriptional shut down in the presence of rDNA breaks. Moreover, we show that failure to inhibit rDNA transcription results in defective rDNA repair, increased genomic instability and reduced cell survival (Fig 7C).

## Discussion

The highly repetitive ribosomal DNA repeats are the most active transcriptional units in the genome. Recently, there has been substantial progress in understanding how rDNA transcription is co-ordinated with rDNA break repair to ensure genomic stability is maintained in the nucleolus (Ciccia *et al*, 2014; Larsen *et al*, 2014; Harding *et al*, 2015; van Sluis & McStay, 2015; Warmerdam *et al*, 2016; Calo *et al*, 2018). Previous studies have shown that in the presence of damage, there is a rapid and transient Pol I silencing that persists in the event of extensive rDNA damage (Kruhlak *et al*, 2007;

van Sluis & McStay, 2015). Our study reports that phosphorylation of H2B on serine 14 is a mechanism of achieving transcriptionally silent nucleolar chromatin in response to damage. We identify a nucleolar fraction of MST2 kinase that binds to nucleolar chromatin and directly phosphorylates H2BS14p upon activation via the ATM-RASSF1A axis. We believe that the specificity of the mark for nucleolar chromatin is due to high presence of the kinase in the nucleolus. Chromatin structure or transcriptional activity may also regulate the establishment of the mark; however, H2BS14p presence in other loci, below the sensitivity of the antibody cannot be excluded. RASSF1A has been shown to get targeted by ATM on serine 131 resulting in increased interaction with MST2 and stimulation of kinase activity (Hamilton *et al*, 2009; Pefani *et al*, 2014). In agreement, we find here that ATM activity is necessary for the establishment of H2BS14p in the nucleolus. Previous studies have shown that ATM regulates Pol II transcription at the sites of damage promoting chromatin remodelling (Shanbhag *et al*, 2010; Kakarougkas *et al*, 2014; Ui *et al*, 2015). Recent observations have highlighted also the importance of ATM in Pol I regulation in response to DNA damage via Nbs1-Treacle- and ARF-dependent interactions (Kruhlak *et al*, 2007; Velimezi *et al*, 2013; Larsen *et al*, 2014; Harding *et al*, 2015). Here, we provide evidence that ATM can also contribute to Pol I DNA damage-dependent rDNA silencing by regulation of nucleolar chromatin architecture via MST2 activation and establishment of H2BS14p. A recent *in vitro* study also showed RASSF1A necessity for the establishment of histone H2B phosphorylation on serine 14, and that a RASSF complexed version of the MST kinase positively regulates the establishment of the mark (Bitra *et al*, 2017).

Phosphorylation of H2BS14 has been reported to promote chromatin condensation both *in vivo* and *in vitro*. It is a mark

**Figure 7.   Loss of nucleolar H2BS14p sensitises cells to rDNA damage.**

A   Clonogenic survival of HeLa or RPE-1 cells that were transfected with the indicated siRNAs and I-PpoI WT or I-PpoI H98A mRNA was introduced after 48 h. The survival ratio I-PpoI WT/I-PpoI H98A in each condition is presented.

B   HeLa cells were transfected with the indicated siRNAs, and 48 h after with I-PpoI WT transcripts, cells were collected at the indicated time points and stained for γH2AX to assess rDNA repair kinetics at the different conditions. Quantification (left) of γH2AX-positive cells and representative images (right) from each condition are shown.

C   Model representing how ATM-dependent nucleolar H2BS14p establishment promotes genomic instability in response to rDNA damage.

Data information: Scale bars at 10 μm. Error bars represent the SD and derive from three independent experiments. Two-tailed Student's *t*-test was used for statistical analysis. *$P < 0.05$, **$P < 0.01$, ***$P < 0.001$.

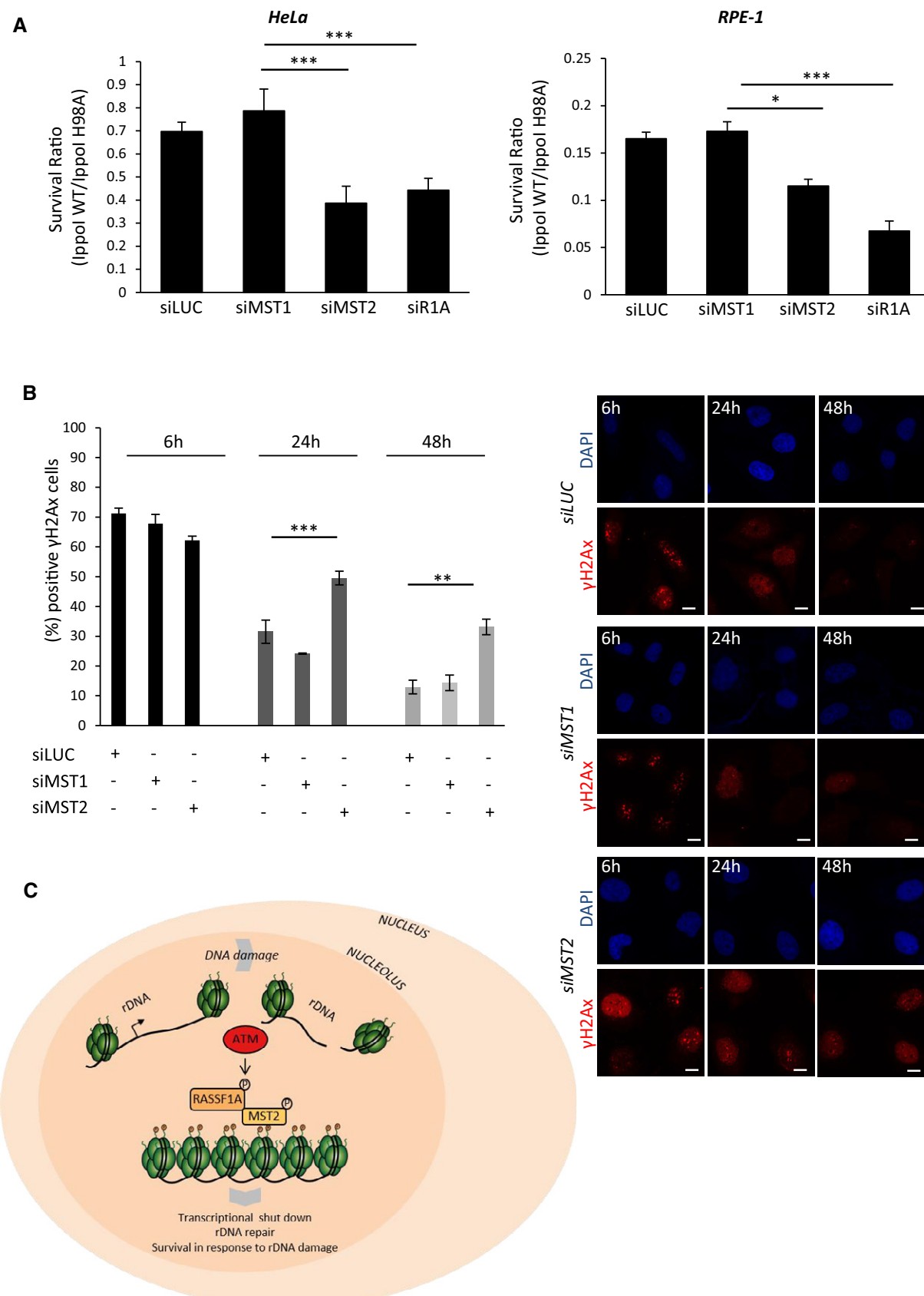

Figure 7.

linked with condensed apoptotic chromatin, highly compacted XY chromosomes as well as at radiation-induced DSB foci late in the DNA damage response (Fernandez-Capetillo *et al*, 2004; Ahn *et al*, 2005). Herein, we show that H2BS14p also marks silent nucleolar chromatin in line with a closed chromatin state to promote chromatin compaction. It was previously shown that RCC1 is a reader of H2BS14p in apoptotic chromatin (Wong *et al*, 2009). In line with that we detected RCC1 recruitment, a factor known to regulate early events in chromatin condensation during mitosis, in the nucleolus with the same kinetics as H2BS14p establishment.

Interestingly, it has been proposed that DNA damage may be differentially regulated in different regions of the nucleus. Repetitive DNA elements such as telomeres, satellite repeats and ribosomal gene arrays pose a unique challenge in DSB recognition and repair due to higher recombinogenic potential compared with the rest of the genome. Epigenetic histone marks dedicated to nucleolar chromatin have been described (Tessarz *et al*, 2014) supporting that chromatin changes in response to DNA damage within the nucleolus could be specific and distinct from most DNA regions.

Recent studies showed that persistent rDNA DSBs move to the exterior of the nucleolus (nucleolar caps) to get access to HR machinery to repair independent of the cell cycle stage (van Sluis & McStay, 2015). The predominant pathway of rDNA breaks though is non-homologous end joining (NHEJ) repair; however, the central components of NHEJ are not found at nucleolar caps (Harding *et al*, 2015). Surprisingly, we did not detect H2BS14p co-localising with γH2Ax at the nucleolar caps; in contrast, we find that the modification marks H2B in the interior of the nucleolus. It is possible that H2BS14p marks a fraction that is repaired by NHEJ or non-damaged nucleolar chromatin that still goes under transcriptional shut down. However, as it has been proposed that the majority of the rDNA moves to the periphery, it is possible that the detection of H2BS14p marks evicted nucleosomes that have been released in the interior to allow efficient repair. Previous studies have shown nucleosome destabilisation around I-PpoI-induced DSBs. This process has been shown to be transient and depend on ATM-Nbs1, both necessary factors for the establishment of rDNA transcriptional shut down upon exposure to damage (Berkovich *et al*, 2007; Goldstein *et al*, 2013).

We previously identified a key role for RASSF1A-MST2 in replication fork stability upon stalling, e.g. in response to DNA breaks, which occurs via activation of LATS1 and promotion of BRCA2-RAD51 nucleofilaments that prevent stalled forks from Mre11 mediated nucleolytic attack (Pefani *et al*, 2014). Here, we further expand these findings to report on RASSF1A-MST2 direct regulation of chromatin and transcriptional repression near sites of DNA breaks, suggesting a wider coordination of chromatin architecture and replication fork stability by the MST2 kinase. Moreover, as MST2 appears enriched in the nucleolus, this suggests that RASSF1A-MST2 signalling may be particularly important in the protection of genomic stability within repetitive elements or highly transcribed areas of the genome.

Previous studies have shown high sensitivity to rDNA breaks. rDNA DSBs produced by the I-PpoI endonuclease significantly affect cell survival depending on the cell type and potentially the degree of oncogenic transformation (Warmerdam *et al*, 2016). Despite variation in sensitivity to I-PpoI between different cell lines, the absence

of nucleolar H2BS14p consistently demonstrated DNA damage and reduced survival. This data highlight Pol I transcriptional shut down within nucleolar chromatin as a necessary measure to allow DNA repair and prevent further damage, in agreement with perturbed rDNA transcription rates being associated with DNA repair defects and genomic instability (Ide *et al*, 2010).

RASSF1A is one of the most commonly epigenetically inactivated genes in human malignancies. RASSF1A CpG island methylation has been shown to correlate with early cancer onset in several tumour types including lung cancer (Grawenda *et al*, 2015; Pefani *et al*, 2016). We and others have shown that RASSF1A inactivation results in genomic instability and increased radio-sensitivity (Dote *et al*, 2005; Yee *et al*, 2012; Pefani *et al*, 2014). Our data provide a new mechanistic insight on how RASSF1A methylation can impact on genomic stability via regulation of MST2 kinase activity and nucleolar chromatin dynamics. Increased rDNA transcription is a common feature of most tumours and a specific target of anticancer therapies (Drygin *et al*, 2010). Therefore, understanding how modifications in nucleolar chromatin can contribute to rDNA silencing is likely to be an important therapeutic avenue in cancer.

## Materials and Methods

### Tissue culture and cell treatments

HeLa, U2OS and RPE1 cells were cultured in complete DMEM supplemented with 10% foetal bovine serum in 5% $CO_2$ and 20% $O_2$ at 37°C. Human bronchial epithelial cells were cultured in keratinocyte serum-free medium supplemented with EGF and bovine pituitary extract (GIBCO). HeLa and U2OS cells were purchased from Cancer Research UK, London, or LGC Promochem (ATCC). HBECS were provided by V.G (Komseli *et al*, 2018). All irradiations were carried out using a Gamma Service® GSRD1 irradiator containing a Cs137 source. The dose rates of the system, as determined by the supplier, were 1.938 Gy/min and 1.233 Gy/min depending on the distance from the source. Cells were exposed in 5 Gy unless stated otherwise. For siRNA transfections, cells were transfected with plasmid DNA (2.5 μg/$10^6$ cells) or siRNA (100 nM) using Lipofectamine 2000 (Invitrogen) according to manufacturer's instructions. I-PpoI WT and I-PpoI H98A mRNA transfections were conducted as previously described (van Sluis & McStay, 2015). In brief, plasmids were linearised at a NotI site and transcribed using the MEGAscript T7 kit (Ambion). I-PpoI mRNA was subsequently polyadenylated using a Poly(A) tailing kit (Ambion) according to the manufacturer's instructions. The *in vitro* transcribed mRNA was transfected using the TransMessenger transfection reagent (Qiagen) according to the manufacturer's instructions. Following 4 h of incubation, the transfection medium was replaced by full medium, and cells were grown for an additional 2 h unless stated otherwise.

### Drug treatments

For ATM inhibition, cells were treated with 10 μM of KU55933, 1 h prior to exposure to γIR or I-PpoI mRNA transfections. For DNA-PK inhibition, cells were with 1 μM NU7441 treated 1 h prior to exposure to γIR. For PARP inhibition, cells were treated with 1 μM

olaparib 2 h prior to exposure to γIR. For polymerase I inhibition, cells were treated with 2 μM CX-5461 for 2 h. To induce apoptotic H2BS14p, cells were treated with 50 μM etoposide overnight.

## 5-EU incorporation

In situ detection of nascent RNA was performed with the Click-iT Alexa Fluor 488 Imaging Kit (Invitrogen, Molecular Probes). Briefly, cells were incubated for the indicated times in the presence of 0.5 mM 5-EU. Samples were fixed in 4% paraformaldehyde for 15 min and permeabilised in 0.5% Triton X-100 for 20 min at room temperature. Samples were then processed according to the manufacturer's recommendation. Cells were analysed using LSM780 or LSM710 (Carl Zeiss Microscopy) confocal microscopes, and 5-EU intensity was quantified with the NIS-elements software (Nikon). At least 100 cells were quantified per condition from at least two independent experiments. For statistical analysis, the Mann–Whitney test was used.

## Immunofluorescence

Cells were grown on coverslips and treated as indicated. Cells were fixed with methanol at −20°C for 10 min, washed with 1× PBS and blocked with 2% BSA in 1× PBS. Coverslips were incubated with the indicated antibodies in blocking solution overnight at 4°C, washed and stained with secondary anti-rabbit and or anti-mouse IgG conjugated with Alexa Fluor secondary antibodies (Molecular Probes) for 1 h at room temperature. Coverslips were washed with PBS + 0.1% Tween, and DNA was stained with DAPI. Cells were analysed using LSM780 (Carl Zeiss Microscopy Ltd) confocal microscope. At least 200 cells were counted from three independent experiments. For detection of H2BS14p, the phospho-histone H2B (Ser14; Cell Signalling, 6959) was used unless stated otherwise.

## Immunoprecipitation and Western blotting

For MST2 immunoprecipitation, cells were treated as indicated and washed with ice-cold PBS prior to lysis. Cells were lysed in 1% NP-40 lysis buffer (150 mM NaCl, 20 mM HEPES, 0.5 mM EDTA) containing complete protease and phosphatase inhibitor cocktail (Roche). Total cell extracts were incubated for 3 h with 20 μl protein A Dynabeads (Invitrogen) and 2 μg of MST2 antibody (ab52641) at 4°C. Total cell extracts (corresponding to 10% of the immunoprecipitate) and immunoprecipitates were resolved in 4–12% Bis–Tris Nu-PAGE gels (Invitrogen) and transferred onto PVDF membrane (Millipore) before immunoblotting with the appropriate antibodies overnight at 4°C. Primary antibody detection was achieved with peroxide-conjugated anti-rabbit or anti-mouse antibodies (Jackson Immunoresearch) and exposure to X-Ray film (Kodak). To quantify the bands obtained with western blot analysis, we used ImageJ software (NIH). All bands were normalised against the loading controls. For detection of H2BS14, the phospho-histone H2B (Ser14; Millipore, 07-191) was used unless stated otherwise.

## Chromatin immunoprecipitation

Chromatin immunoprecipitation from U2OS cells was performed with the Chip-IT High Sensitivity kit (Active Motif) according to manufacturer's instructions. In brief, U2OS cells were exposed or not to 5 Gy

of γIR and cells fixed 30 min after. Samples were sonicated for 40 cycles 30 s ON/OFF on the Bioruptor (Diagenode). Inputs were normalised on their content, and 30 μg of chromatin/condition was immunoprecipitated using 4 μg of MST2 (ab52641) or rIgG isotype control (#3900). Samples were eluted and de-crosslinked. DNA was analysed by qPCR. Fold enrichment over IgG control was calculated using the $2^{(\Delta\Delta C_t)}$ method (all primers used are listed below).

## Quantitative real-time PCR analysis

RNA extraction, reverse transcription and qPCR were performed using the Ambion® PowerSYBR® Green Cells-to-CT™ kit following manufacturer's instructions in a 7500 FAST Real-Time PCR thermocycler with v2.0.5 software (Applied Biosystems). mRNA fold change was calculated using a $2^{(\Delta\Delta C_t)}$ method in relation to GAPDH reference gene (all primers used are listed below).

## Cellular fractionation

Cellular fractionation was performed as described elsewhere (Ahmad et al, 2009). In brief, cells were trypsinised, pelleted and washed with 1× PBS. Cell pellets were resuspended in hypotonic buffer and incubated on ice for 10 min. Cells were broken open to release nuclei with a Dounce homogeniser. Nuclei were isolated after centrifuging over sucrose cushions. Nucleoli were released from the purified nuclei by sonication and isolated with centrifuging over sucrose cushions.

## Clonogenic survival assay

Cells were treated with siRNAs and 48 h post-transfection I-Ppo I WT, or I-PpoI H98A mRNA was introduced TransMessenger transfection reagent (Qiagen). 6 h post-mRNA transfection, cells were counted, and 800 cells were seeded onto 60-mm-diameter dishes and incubated for 10–14 days in regular medium. Plates were stained with crystal violet (0.5% w/v crystal violet, 50% v/v MeOH and 10% v/v EtOH). Experiments were performed in triplicates performed at least two independent times as stated in figure legends.

## Plasmids

I-PpoI WT and I-PpoI H98A were kindly provided by Brian Mc Stay (Galway University); H2B-GFP and H2BS14D-GFP were kindly provided by Silvia Soddu (IFO).

## Antibodies

MST2 (Abcam, ab52641), phospho-histone H2B (Ser14) (Cell Signalling, 6959), phospho-histone H2B (Ser14) (Millipore, 07-191), MST1 (Millipore, 07-061), RASSF1A (3F3, Santa Cruz sc-58470), UBF (F9, Santa Cruz, sc-13125), SAV-1 (Atlas, HPA001808), V5 (Cell Signalling, 13202), H2B (abcam, ab52484), H2A (Abcam, ab18255), H3 (96C10, Cell Signalling, 3638), H4 (Cell Signalling, 2592), γH2AX (JBW301, Millipore, 16-193), nucleolin (4E2, Abcam, ab13541), lamin A/C (Cell signalling, 4777), α-tubulin (B3, Sigma, T9822), RCC1 (Cell Signalling, 5134), phospho-MST1 (Thr183)/MST2 (Thr180) (Cell Signalling, 3681), KAP1 (Bethyl, A300-274A) and phospho-KAP1S824 (Cell signalling, 4127).

**Oligo sequences**

Real-time PCR primers:
Pre-rRNA (FW): 5′ CCGCGCTCTACCTTACCTAC 3′
Pre-rRNA (REV): 5′ GAGCGACCAAAGGAACCATA 3′
GAPDH (FW): 5′ ATCCCATCACCATCTTCCA 3′
GAPDH (REV): 5′ GGACTCCACGACGTACTCA 3′
B2M (FW): 5′ CTCCGTGGCCTTAGCTGTG 3′
B2M (REV): 5′ TTTGGAGTACGCTGGATAGCCT 3′
ChIP primers:
H0 (FW): 5′ GGTATATCTTTCGCTCCGAG 3′
H0 (REV): 5′ GACGACAGGTCGCCAGAGGA 3′
H1 (FW): 5′ GGCGGTTTGAGTGAGACGAGA 3′
H1 (REV): 5′ ACGTGCGCTCACCGAGAGCAG 3′
H18 (FW): 5′ GTTGACGTACAGGGTGGACTG 3′
H18 (REV): 5′ GGAAGTTGTCTTCACGCCTGA 3′
GAPDH (FW): 5′ TACTAGCGGTTTTACGGGCG 3′
GAPDH (REV): 5′ TCGAACAGGAGGAGCAGAGAGCG 3′

   siRNA oligos: luciferase: GCCAUUCUAUCCUCUAGAGGAUG, siMST2: siGENOME smartpool: M-004874-02 (Dharmacon), siMST2_2: HSS110314 (Invitrogen), siMST1: GGGCACUGUCCGA-GUAGCA, siRASSF1A: GACCUCUGUGGCGACUUCA and siSAV1: GCACAUGAAGACUACAG.

**Expanded View** for this article is available online.

## Acknowledgements
We would like to thank Brian McStay for providing reagents and for helpful discussions.

## Author contributions
DEP and EON conceived the project and designed the research. DEP, MLT and DPE performed and analysed experiments. VG offered reagents and scientific advice. DEP and EON wrote the manuscript.

## Conflict of interest
The authors declare that they have no conflict of interest.

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
