## [Review Process File · The EMBO Journal]

MST2 kinase suppresses rDNA transcription in response to DNA damage by phosphorylating nucleolar histone H2B

Dafni Eleftheria Pefani, Maria Laura Tognoli, Deniz Pirincci Ercan, Vassilis Gorgoulis and Eric O'Neill

Review timeline:

Submission date:	4 December 2017
Editorial Decision:	21 December 2017
Revision received:	21 March 2018
Editorial Decision:	9 April 2018
Revision received:	18 April 2018
Accepted:	27 April 2018

Editor: Hartmut Vodermaier

Transaction Report:

1st Editorial Decision

21 December 2017

Thank you for submitting your manuscript on H2B-S14 phosphorylation in the nucleolar DNA damage response for our editorial consideration. We have now received the enclosed reports from three expert referees, in light of which we should be happy to consider a revised version of this manuscript further for publication. Nevertheless, despite their overall support for the study, all reviewers point out a number of important issues that will need to be decisively addressed before EMBO Journal publication would be warranted. Among the recurrent key concerns are the request for more definitive epistasis experiments (utilizing also H2B S14A substitutions), and the need for various additional controls, increased sample sizes and additional examples, as well as quantification of results.

I would thus like to invite you to address the referees' comments by way of a revised version of the manuscript. Please keep in mind that our policy to allow only a single round of major revision will make it important to carefully answer to all points raised at this stage - therefore, please do not hesitate to get back to me with any questions/comments you may have regarding the referee reports already during the early stages of your revision. We might further discuss possible extension of the revision period (beyond the regular three months), during which time the publication of any competing work elsewhere would have no negative impact on our final assessment of your own study.

Please refer to the sections below for additional information on preparing and uploading a revised manuscript.

Thank you again for the opportunity to consider this work for The EMBO Journal, and I look forward to hearing from you in due time.

REFeree REPORTS

Referee #1:

This manuscript by O'Neill and colleagues provides a compelling dissection of the molecular basis for DNA damage-induced rRNA silencing. The work is overall well-presented and convincing. While the use of the H2BS14D mutant supports a direct role for H2BS14 phosphorylation in rRNA repression, it would be important to complement these studies with an S14A mutant that cannot be phosphorylated and should therefore impair damage-induced silencing.

Minor comments:

- 1) The images for MST1 and MST2 in Fig 2B are not very convincing, much better images for MST2 nucleolar localization are shown in subsequent panels.
- 2) More than one housekeeping gene (GAPDH is listed) should be used for normalization of pre-mRNA expression, particularly when overexpressing histone mutants.
- 3) Does the loss of H2B phosphorylation upon MST2/RASSF1A depletion reflect a reduction or a kinetic delay?
- 4) I-PpoI was recently shown to cut degenerate recognition sites in mouse cells, resulting in an estimated > 100 non-nucleolar DSB sites (PMID: 26687720).
- 5) How does the observed effect on rRNA silencing relate to RASSF1A/MST2 function during replication stress, reported previously by the same group?

Referee #2:

Pefani et al. report on the involvement of the histone modification H2BS14p in transcriptional repression within nucleoli upon DNA damage. Previously this post-translational modification has primarily been associated with apoptotic signalling, during which the kinase MST1 is responsible for establishment of this modification. In this study, the authors make the interesting observation that after DNA damage the paralogue MST2 is responsible for transient nucleolar H2BS14p and that this is necessary for damage-induced transcriptional repression. The authors provide evidence that MST2 activation occurs downstream of the DDR kinase ATM and is mediated by the scaffold protein and ATM target RASSF1A. Further, the authors show that cells lacking MST2 or RASSF1A, which results in impaired H2BS14p and failure to suppress nucleolar transcription, have compromised genome stability and cell survival upon rDNA damage. The identification of a new role for H2BS14p in the nucleolar DNA damage response is exciting and novel, and the dynamic phosphorylation of this mark may have important functions for nucleolar integrity and rDNA stability. However, additional experiments, important controls and quantification of imaging data would be needed to strengthen the main conclusions and fully corroborate the study. Specifically, the following points should be addressed:

Major points

1. While the authors provide multiple lines of evidence to link the triad of ATM, RASSF1A and MST2 to nucleolar transcriptional repression upon DNA damage, the only experiment to directly demonstrate that this works through H2BS14p is over-expression of an H2BS14D mutant, which results in decreased Pol I transcription. The specificity of this effect should be controlled employing a phospho-deficient H2B mutant (e.g. H2BS14A, which was used successfully before), and expression levels of GFP-histones should be quantified to ensure equal expression.
2. The same phospho-deficient H2B mutant could serve as a crucial specificity control for the two H2BS14p antibodies used in IF experiments. Further, it would be important to indicate which of the two antibodies has been used for which experiment.
3. From a conceptual point of view, how do the authors envisage that H2BS14 phosphorylation leads to repression of Pol I transcription? The model depicts increased nucleosome density,

presumably to indicate compaction, but this is not demonstrated by the experiments. In fact, it remains completely unclear how H2BS14p is linked to repression. Would readers of this mark (e.g. RCC1) be involved? While a complete dissection of the molecular mechanism downstream of H2BS14p might go beyond the scope of this study, it would be helpful to test obvious candidates and discuss potential mechanisms.

4. Most experiments were performed with a single siRNA against MST2. Key results, e.g. on the transcriptional repression upon rDNA breaks after IR and I-PpoI should be repeated with a second, independent siRNA. Further, could the authors perform rescue experiments with wild-type MST2 and a catalytic mutant to strengthen their findings?
5. Besides ATM also DNA-PK (Pankotai et al. 2012) and PARP activity (Chou et al. 2010, Awwad et al. 2017) have been linked to transcriptional repression after DNA damage. It would be informative if the authors could test whether DNA-PK and PARP inhibition would play a role in nucleolar transcription shut-off in their systems, especially since ATM inhibition does not seem to fully rescue rDNA transcription (Fig. 3B).
6. Along the same lines, epistasis experiments between MST2 knockdown and ATM inhibition would strengthen the conclusions on their cooperation for damage-induced rDNA repression.
7. To more directly link ATM to MST2 activation, it would be useful to assess changes in MST2 activation/phosphorylation upon IR and ATM inhibition by Western Blot.
8. The comparability of the time-course experiments in the manuscript suffers a bit from being done at different time-points for different experiments. This makes it challenging to judge if e.g. the transcriptional shutdown (3A, 10min, 30min) follows similar kinetics as the H2BS14p modification (Fig 1A, B, C, D, 30min, 1h, 2h or 10min, 20min, 40min). Similarly, does e.g. EU incorporation follow inverse kinetics as the H2BS14p modification?
9. In Figure 2E the authors show increased interaction between MST2 and H2B after IR. Is this specific for H2B? How would H2A and H3/H4 look in this interaction assay? If the increased histone interaction of MST2 after damage was a general phenomenon that happens on both nucleolar and non-nucleolar chromatin, why would only nucleolar H2B become phosphorylated?
10. In several cases single cell images are shown and quantifications from larger cell cohorts would be very beneficial. Where quantifications are provided, it should be indicated what the error bars signify and not just how many experiments they are derived from. Further, it would be useful to provide information on the number of cells that were analysed. Significance levels (*, **, ***, etc.) should be explained in the figure legends and it should be made clear which two samples were compared against each other.

Minor points:

1. In Figure 1D it would be useful to have time-points beyond 40min and/or similar to the time-points taken for the immunofluorescence to fully support the 'similar kinetics'.
2. In EV1C, at the 10min time-point the DAPI channel is not included in the merged image and the cell images in EV1C after 1h and 2h are identical.
3. In Figure 5, it would help corroborate the relation between the rDNA damage and the H2BS14p modification by a Western Blot showing (presumably) increase in the modification levels with similar kinetics as the gH2AX signal in the nucleolus after transfection of the V5-I-PpoI.
4. In Figure 2D, why is the merged image not shown for this panel, when it is for all other ones?
5. The different channels and the merge in Figure 4A (and throughout) should be properly aligned, the y-axis seems off in some images relative to others and the same is probably the case for the x-axis, although less easy to spot.

Referee #3:

In this manuscript Eric O'Neill and colleagues discover that histone H2B phosphorylation on Ser14 specifically marks damaged rDNA repeats in the nucleoli in response to treatments that induce DNA double-strand breaks (DSBs). They demonstrate that H2B is phosphorylated on Ser14 in the nucleoli in response to ionizing radiation (IR) treatment in an ATM-dependent manner. They then present convincing evidence that the MST2 kinase phosphorylates H2B in response to IR. Furthermore, the authors show that H2B Ser14 phosphorylation is associated with rRNA transcriptional suppression in response to DNA damage and that this chromatin mark is somehow contributing to the rRNA transcriptional repression and to survival of cells upon rDNA breakage. Based on these data they propose that ATM-dependent activation of MST2 specifically marks histone H2B by phosphorylation in rDNA chromatin, leading to transcriptional repression of the rRNA gene clusters in the nucleoli, which may facilitate DSB repair in these highly transcribed and repetitive chromatin regions.

This is an interesting paper. Even though the data set is somewhat descriptive and short of mechanistic insights, I still think it is interesting and novel enough to warrant publication in EMBO J. I have a few suggestions for additional experiments that would further strengthen this story.

1) On several occasions key findings are presented by fluorescence microscopy micrographs where just one cell is shown. In my opinion the paper would benefit from some quantification of the microscopy experiments.

2) It was previously shown that nucleolar segregation in response to DNA damage and/or Pol I inhibition is associated with movement of the rDNA repeats from the nucleolar interior into nucleolar caps (e.g. van Sluis and McStay 2015). Surprisingly, judging from Figure 5, this relocation does not seem to happen for the pS14 H2B signal. Do we have to conclude that H2B remains in the nucleoli while the rDNA repeats are localized to the caps? And does this mean that it is free H2B that is modified on Ser14 by MST2? This should perhaps be analyzed in more detail and/or discussed in the text.

3) It is still unclear if transcriptional repression of the rDNA repeats is a cause or a consequence of nucleolar segregation. Therefore it would be interesting to test if MST2 silencing is also affecting nucleolar segregation (cap formation).

Minor points:

1) Manuscript pages should be numbered, otherwise it's difficult to refer to certain passages of the text

2) Somewhere towards the end of the paper, there is this sentence: "Nucleolar caps are formed under these conditions due to the vast levels of breaks in the nucleoli (van Sluis & McStay, 2015)". This is not entirely correct. Nucleolar caps also form under conditions where few breaks are induced in the rDNA repeats (i.e. upon IR treatment) (see Kruhlak et al., 2007). However, in response to IR, fewer caps form and they are not present in all the nucleoli. Thus, I-Ppo1 treatment just accents this physiological response. This should be clarified in the text.

We would like to thank the reviewers for their helpful and constructive comments. In our revised manuscript we have now addressed their concerns with additional experimental data that further supports our model. Please find below a point by point reply to the raised comments:

Referee #1:

This manuscript by O'Neill and colleagues provides a compelling dissection of the molecular basis for DNA damage-induced rRNA silencing. The work is overall well-presented and convincing. While the use of the H2BS14D mutant supports a direct role for H2BS14 phosphorylation in rRNA repression, it would be important to complement these studies with an S14A mutant that cannot be phosphorylated and should therefore impair damage-induced silencing.

We would like to thank the reviewer for appreciating the novelty of our work and supporting publication in EMBO J. We have now performed additional experiments using the H2BS14A-GFP phospho-deficient mutant as suggested. These new experiments indeed demonstrate the impact of the H2BS14A mutant on rDNA transcription upon induction of damage with γ IR and the I-Ppo I endonuclease is reciprocal to the phospho-mimetic mutant H2BS14D (Fig 4E, EV3A, EV3D) and in keeping with our model.

Minor comments:

1) The images for MST1 and MST2 in Fig 2B are not very convincing, much better images for MST2 nucleolar localization are shown in subsequent panels.

We now provide new images for Fig 2B as well as lower magnification images displaying intracellular MST1 and MST2 staining in multiple cells in Figure 2A and Figure 2D.

2) More than one housekeeping gene (GAPDH is listed) should be used for normalization of pre-mRNA expression, particularly when overexpressing histone mutants.

We now included additional normalisation of pre-rRNA expression against beta-2-microglobulin as well as GAPDH for all the experiments where we have overexpressed histone variants. New data presented in Figures EV3C, EV3D, EV4G.

3) Does the loss of H2B phosphorylation upon MST2/RASSF1A depletion reflect a reduction or a kinetic delay?

We now show data from time points up to 4 hours post exposure to γ IR from cells that have been treated with siRNA against MST2 compared to control siRNA (Fig EV2D). As we were not able to detect H2BS14p in any of the timepoints in the absence of MST2, we suggest that MST2 depletion results in a reduction rather than a kinetic delay in the H2BS14p establishment.

4) I-Ppol was recently shown to cut degenerate recognition sites in mouse cells, resulting in an estimated > 100 non-nucleolar DSB sites (PMID: 26687720).

This is correct; however the human genome only contains 13 canonical sites in non-nucleolar DNA (Muscarella et al, 1990) and one site located within the 28S rDNA coding region within the repetitive 45S rDNA locus. Degenerate DNA recognition has also been reported (Wittmayer et al., 1998) but as the 45S rDNA locus has > 300 repeats this represents the majority of specific sites in the genome and responsible for high levels of DSB formation within the nucleolus. Indeed the majority of γ H2Ax staining after introduction of the I-Ppol mRNA is evident at the nucleolar caps (Fig 6A), therefore we believe this system provides an appropriate model for an enriched nucleolar DNA damage response as it has also been described previously (Van Sluis and McStay, 2015, Harding et al., 2015, Warmerdam et al. 2016).

5) *How does the observed effect on rRNA silencing relate to RASSF1A/MST2 function during replication stress, reported previously by the same group?*

In Pefani et al. (2014) we presented a model for ATR-RASSF1A-MST2-LATS1 that maintained genome stability through protection of stalled forks. Here we report an ATM-RASSF1A-MST2 mediated regulation of chromatin in response to DSBs in rDNA at the nucleolus. We believe that these are two stimuli that are required to resolve breaks and prevent increased replication defects. The high concentration of MST2 at the nucleolus suggests that our original finding may have greater relevance to rDNA and would correlate with emerging evidence for rDNA damage being more detrimental due to the extensive repetitive nature of the locus (Ide et al, 2010, Warmerdam et al, 2016). This is something we are now addressing in detail and have included the following paragraph in the discussion (p8).

'We previously identified a key role for RASSF1A-MST2 in replication fork stability upon stalling, e.g. in response to DNA breaks, which occurs via activation of LATS1 and promotion of BRCA2-RAD51 nucleofilaments that prevent stalled forks from Mre11 mediated nucleolytic attack. Here, we further expand these findings to report on RASSF1A-MST2 direct regulation of chromatin and transcriptional repression near sites of DNA breaks, suggesting a wider coordination of chromatin architecture and replication fork stability by the MST2 kinase. Moreover, as MST2 appears predominantly localised in the nucleolus this suggests that RASSF1A-MST2 signalling may be particularly important in the protection of genomic stability within repetitive elements or highly transcribed areas of the genome.'

Referee #2:

Pefani et al. report on the involvement of the histone modification H2BS14p in transcriptional repression within nucleoli upon DNA damage. Previously this post-translational modification has primarily been associated with apoptotic signalling, during which the kinase MST1 is responsible for establishment of this modification. In this study, the authors make the interesting observation that after DNA damage the paralogue MST2 is responsible for transient nucleolar H2BS14p and that this

is necessary for damage-induced transcriptional repression. The authors provide evidence that MST2 activation occurs downstream of the DDR kinase ATM and is mediated by the scaffold protein and ATM target RASSF1A. Further, the authors show that cells lacking MST2 or RASSF1A, which results in impaired H2BS14p and failure to suppress nucleolar transcription, have compromised genome stability and cell survival upon rDNA damage. The identification of a new role for H2BS14p in the nucleolar DNA damage response is exciting and novel, and the dynamic phosphorylation of this mark may have important functions for nucleolar integrity and rDNA stability. However, additional experiments, important controls and quantification of imaging data would be needed to strengthen the main conclusions and fully corroborate the study. Specifically, the following points should be addressed:

We would like to thank the reviewer for recognising the novelty of our work. Below there is a detailed description in all the experiments and changes performed to answer their concerns.

Major points

1. While the authors provide multiple lines of evidence to link the triad of ATM, RASSF1A and MST2 to nucleolar transcriptional repression upon DNA damage, the only experiment to directly demonstrate that this works through H2BS14p is over-expression of an H2BS14D mutant, which results in decreased Pol I transcription. The specificity of this effect should be controlled employing a phospho-deficient H2B mutant (e.g. H2BS14A, which was used successfully before), and expression levels of GFP-histones should be quantified to ensure equal expression.

We have now performed the experiments using the H2BS14A-GFP phospho-deficient mutant. Expression levels and intracellular expression pattern of H2B-GFP, H2BS14A-GFP and H2BS14D-GFP is now shown in Figure EV3A. We now provide data that assess the impact of the H2BS14A mutant in rDNA transcription upon induction of damage with γ IR and the I-PpoI endonuclease (Fig 4E, EV3D). In both cases we observed increased abundance of pre-rRNA transcripts in cells transfected with the H2BS14A-GFP variant compared to control cells.

2. The same phospho-deficient H2B mutant could serve as a crucial specificity control for the two H2BS14p antibodies used in IF experiments. Further, it would be important to indicate which of the two antibodies has been used for which experiment.

We now provide this data in Figure EV1B. Information of the antibody used in each experiment is provided in material and methods. For most of the immunofluorescence experiments the H2BS14p antibody from cell signalling was utilised unless stated otherwise.

3. From a conceptual point of view, how do the authors envisage that H2BS14 phosphorylation leads to repression of Pol I transcription? The model depicts increased nucleosome density, presumably to indicate compaction, but this is not demonstrated by the experiments. In fact, it remains completely unclear how H2BS14p is linked to repression. Would readers of this mark (e.g. RCC1) be involved? While a complete dissection of the molecular mechanism downstream of H2BS14p might go beyond

the scope of this study, it would be helpful to test obvious candidates and discuss potential mechanisms.

We agree with the reviewer that a direct experimental validation of our hypothesis that the H2BS14p mark results in chromatin condensation in the nucleolus is missing. This is due to technical reasons as a Micrococcal Nuclease (MNase) assay in isolated nucleoli has proven technically challenging. However, there is a well-established connection between H2BS14p and chromatin compaction both *in vitro* and *in vivo* (Cheung et al, 2003; de la Barre et al, 2001). As suggested, we have now looked for RCC1 as a possible reader of nucleolar H2BS14p. Indeed, we observed accumulation of RCC1 in the nucleoli with the same kinetics as the establishment of H2BS14p response to γ IR. This new data are now shown in Figure 4F. This finding supports our hypothesis that nucleolar H2BS14p would lead in nucleosome density changes and opens interesting avenues on how DNA damage can regulate RAN-GTP gradient and nuclear import-export that immobilisation of RCC1 onto chromatin was shown to regulate before (Wong et al, 2009), that however is beyond the scope of the current manuscript.

4. Most experiments were performed with a single siRNA against MST2. Key results, e.g. on the transcriptional repression upon rDNA breaks after IR and I-Ppol should be repeated with a second, independent siRNA. Further, could the authors perform rescue experiments with wild-type MST2 and a catalytic mutant to strengthen their findings?

We have observed that overexpression of MST2 or the catalytically inactive mutant MST2K56R is highly enriched in the cytoplasm and the nuclear fraction is very low. Therefore, we believe that overexpression experiments cannot be confidently interpreted. However, we now provide a full analysis on the impact of MST2 on rDNA transcription using a second siRNA oligo against MST2 (siMST2_2). In the original version of our MS we showed that siMST2_2 also results in loss of H2BS14p (Figure EV2H). We now show siMST2_2 also results in increased pre-rRNA transcription in response to γ IR (Figure EV2I and EV2J). We also tested the second siRNA oligo to assess incorporation of 5-EU upon nucleolar DSB formation using the I-Ppol endonuclease (Figure EV4F). The data with siMST2_2 are similar to the data obtained using the original smart pool supporting the specificity of our phenotype.

5. Besides ATM also DNA-PK (Pankotai et al. 2012) and PARP activity (Chou et al. 2010, Awwad et al. 2017) have been linked to transcriptional repression after DNA damage. It would be informative if the authors could test whether DNA-PK and PARP inhibition would play a role in nucleolar transcription shut-off in their systems, especially since ATM inhibition does not seem to fully rescue rDNA transcription (Fig. 3B).

In agreement with previous publications (Hamilton et al., 2009), in newly added data presented in Figure 5B we detect high levels of pMST2 in the presence of γ IR and this is significantly affected by inhibition of ATM kinase. To test whether PARP or DNA-PK regulate MST2 kinase activity we looked for MST2 auto-phosphorylation and subsequent activation under conditions that PARP or DNA-PK

activity are inhibited. In contrast to ATM inhibition, we now show inhibition of DNA-PK or PARP does not result in changes in MST2 kinase activity (Fig EV3F). However, it is important to point out that this data does not exclude the possibility that DNA-PK or PARP can impact on rDNA transcription in response to γ IR via MST2 kinase activity independent routes (Calkins et al, 2013).

6. Along the same lines, epistasis experiments between MST2 knockdown and ATM inhibition would strengthen the conclusions on their cooperation for damage-induced rDNA repression.

We now provide new qPCR data that show that in agreement with ATM acting upstream of MST2 we do not see any additive impact on rDNA transcription upon concomitant depletion of MST2 expression with an siRNA and inhibition of ATM (Fig 5C). We did observe though higher levels of pre-rRNA transcripts in the presence of ATM inhibition compared to siMST2 deletion which suggests that ATM also impacts on rDNA transcription independently of MST2 activation (Larsen et al., 2014, Ciccia et al., 2014). This is also suggested by the data on 5-EU incorporation (Fig 6F, Fig EV4B) upon rDNA DSB formation with the I-Ppol that ATM inhibition has a more profound impact than MST2 knockdown on the rescue of Pol I transcription.

7. To more directly link ATM to MST2 activation, it would be useful to assess changes in MST2 activation/phosphorylation upon IR and ATM inhibition by Western Blot.

This data is now provided as Fig 5B. We show increased phosphorylation of MST2 at Thr 180 upon exposure to γ IR that depends on ATM kinase inhibition in agreement with previous studies (Hamilton et al, 2009, Yee et al., 2012).

8. The comparability of the time-course experiments in the manuscript suffers a bit from being done at different time-points for different experiments. This makes it challenging to judge if e.g. the transcriptional shutdown (3A, 10min, 30min) follows similar kinetics as the H2BS14p modification (Fig 1A, B, C, D, 30min, 1h, 2h or 10min, 20min, 40min). Similarly, does e.g. EU incorporation follow inverse kinetics as the H2BS14p modification?

We would like to point out that all our experiments in response to γ IR are conducted within a timeline of 40 mins that has been shown by us here and others to be the timeframe in which Pol I transcriptional shut down takes place in response to γ IR (Fig 4A, Kruhlak et al, 2007; Larsen et al, 2014). This is in agreement with the time that we detect nucleolar H2BS14p (Fig 1C and 1D).

Polymerase I inhibition in response to γ IR is a transient event (Kruhlak et al, 2007; Larsen et al, 2014). We also hypothesize that the levels or kinetics of Pol I inhibition may be affected by the position of the break and/or the amount of damage in each cell. Since we are using different methods to assess the establishment of H2BS14p and Pol I transcription we had to stratify our time points to overcome technical difficulties (e.g. 10 min exposure to 5-EU are not sufficient for a detectable signal). 20 min after exposure to γ IR we can detect MST2 dependent accumulation of H2BS14p in the nucleolus (Fig 3C and 3D) and reduced rDNA transcription that can be significantly rescued by MST2 depletion as assessed both by qPCR and 5-EU incorporation experiments (Fig 4B and 4C). We also show the inverse kinetics between H2BS14p and 5-EU incorporation in response to DSBs formed with the I-Ppol endonuclease (Fig EV4C and EV4D).

9. In Figure 2E the authors show increased interaction between MST2 and H2B after IR. Is this specific for H2B? How would H2A and H3/H4 look in this interaction assay?

We have now looked for H2A, H3 and H4 in MST2 immunoprecipitates in the presence and absence of DNA damage and identified increased presence upon exposure to γ IR, indicating that MST2 does not bind to free H2B pools but with chromatin bound nucleosomes (Fig EV4B). Additionally, we have now performed CHIP-qPCR experiments to assess MST2 binding to the rDNA locus and detected MST2 across the DNA repeat soon after exposure to γ IR (Fig 3B), suggesting association with the rDNA repeats.

If the increased histone interaction of MST2 after damage was a general phenomenon that happens on both nucleolar and non-nucleolar chromatin, why would only nucleolar H2B become phosphorylated?

Apologies if we were misleading in the original submission, we believe this is apparent at the rDNA repeats as MST2 is enriched at the nucleolus and binds to nucleolar chromatin upon DNA damage, as we now show in CHIP-qPCR experiments (Fig 3B). This may occur at other loci but below detection potentially due to absence of MST2, lower levels of active transcription or different chromatin structure. We clarify and discuss this in our MS (p.7).

10. In several cases single cell images are shown and quantifications from larger cell cohorts would be very beneficial.

We now include lower magnification images and quantified data for all our key experiments. Additional quantifications are given in Figures 3C, 4F, 5A, 5D, EV4B.

Where quantifications are provided, it should be indicated what the error bars signify and not just how many experiments they are derived from. Further, it would be useful to provide information on the number of cells that were analysed. Significance levels (, **, ***, etc.) should be explained in the figure legends and it should be made clear which two samples were compared against each other.*

We have now added this info in figure legends and the material and methods section.

Minor points:

1. In Figure 1D it would be useful to have time-points beyond 40min and/or similar to the time-points taken for the immunofluorescence to fully support the 'similar kinetics'.

We removed the phrase similar kinetics.

2. In EV1C, at the 10min time-point the DAPI channel is not included in the merged image and the cell images in EV1C after 1h and 2h are identical.

We have now included the DAPI channel at the 10 min time point and corrected the 2h time point image.

3. In Figure 5, it would help corroborate the relation between the rDNA damage and the H2BS14p modification by a Western Blot showing (presumably) increase in the modification levels with similar kinetics as the gH2AX signal in the nucleolus after transfection of the V5-I-Ppol.

We now provide this data (Fig 6A).

4. In Figure 2D, why is the merged image not shown for this panel, when it is for all other ones?

Merged channels are now shown.

5. The different channels and the merge in Figure 4A (and throughout) should be properly aligned, the y-axis seems off in some images relative to others and the same is probably the case for the x-axis, although less easy to spot.

We have now carefully aligned all our images.

Referee #3:

In this manuscript Eric O'Neill and colleagues discover that histone H2B phosphorylation on Ser14 specifically marks damaged rDNA repeats in the nucleoli in response to treatments that induce DNA double-strand breaks (DSBs). They demonstrate that H2B is phosphorylated on Ser14 in the nucleoli in response to ionizing radiation (IR) treatment in an ATM-dependent manner. They then present convincing evidence that the MST2 kinase phosphorylates H2B in response to IR. Furthermore, the authors show that H2B Ser14 phosphorylation is associated with rRNA transcriptional suppression in response to DNA damage and that this chromatin mark is somehow contributing to the rRNA transcriptional repression and to survival of cells upon rDNA breakage. Based on these data they propose that ATM-dependent activation of MST2 specifically marks histone H2B by phosphorylation in rDNA chromatin, leading to transcriptional repression of the rRNA gene clusters in the nucleoli, which may facilitate DSB repair in these highly transcribed and repetitive chromatin regions.

This is an interesting paper. Even though the data set is somewhat descriptive and short of mechanistic insights, I still think it is interesting and novel enough to warrant publication in EMBO J. I have a few suggestions for additional experiments that would further strengthen this story.

We thank the reviewer for their comments that our manuscript sufficiently interesting for publication in EMBO J. While the technical challenges of addressing the specific mechanism in a minor fraction of the genome are difficult, we have attempted to answer all comments and now provide additional data that we feel does address this referee's concerns.

1) On several occasions key findings are presented by fluorescence microscopy micrographs where just one cell is shown. In my opinion the paper would benefit from some quantification of the microscopy experiments.

We now show lower magnification images of multiple cells and provide quantified data for all our key experiments. Additional quantifications are given in Figures 3C, 4G, 5A, 5D, 6F and EV4B.

2) It was previously shown that nucleolar segregation in response to DNA damage and/or Pol I inhibition is associated with movement of the rDNA repeats from the nucleolar interior into nucleolar caps (e.g. van Sluis and McStay 2015). Surprisingly, judging from Figure 5, this relocation does not seem to happen for the pS14 H2B signal. Do we have to conclude that H2B remains in the nucleoli while the rDNA repeats are localized to the caps? And does this mean that it is free H2B that is modified on Ser14 by MST2? This should perhaps be analyzed in more detail and/or discussed in the text.

Two recent studies from van Sluis & McStay and Harding et al. offer a very detailed characterisation on how DSBs in the rDNA repeats impact on Pol I inhibition, nucleolar re-organisation and how their repair is achieved. In these studies it was shown that damaged rDNA is re-localised in the nucleolar periphery where it gets repaired by Homologous Recombination (HR). In Harding et al. Non-Homologous End Joining (NHEJ) was also found to have a central role in rDNA DSB repair. However central components of the NHEJ machinery (DNA-PK, KU70/80) do not re-localise in the nucleolar caps suggesting that NHEJ takes place in the nucleolar interior potentially with different kinetics from HR.

van Sluis & McStay, reported that at 6h post I-Ppol mRNA transfection only 20% of rDNA repeats acquire a DSB (van Sluis & McStay, 2015). The complete lack of 5-EU incorporation under these conditions (Fig 6B and EV4B, van Sluis & McStay, 2015) suggests that the transcriptional shut down must also take place within the rDNA repeats that are not damaged (i.e. the other 80%).

We now provide CHIP-qPCR experiments and co-immunoprecipitation experiments to assess MST2 interaction with all the core histones of the nucleosome within the rDNA repeats (Fig 3B and EV2B), that support that MST2 should modify H2B on nucleolar chromatin rather than free H2B pools. Surprisingly though we did not detect H2BS14p co-localising with γ H2Ax at the nucleolar caps, in contrast we show now data that the modification marks H2B in the interior of the nucleolus (Fig EV5B). While it is possible that H2BS14p marks a fraction that is repaired by NHEJ or the non-damaged nucleolar chromatin that is transcriptionally shut down *in trans*, it has been proposed that in the presence of rDNA breaks the majority of the rDNA moves to the periphery (van Sluis & McStay, 2015). Therefore, we may detect H2BS14p in evicted nucleosomes that have been released in the interior to allow efficient repair in the cap. Previous studies have shown nucleosome destabilisation is required around I-Ppol induced DSBs to allow efficient repair. This process has been shown to be transient and depend on ATM-Nbs1, both necessary factors for the establishment of rDNA transcriptional shut down (Berkovich et al, 2007; Goldstein et al, 2013). These points are now discussed in page 8 of the MS.

3) It is still unclear if transcriptional repression of the rDNA repeats is a cause or a consequence of nucleolar segregation. Therefore it would be interesting to test if MST2 silencing is also affecting nucleolar segregation (cap formation).

We have now looked for nucleolar segregation in the presence of I-Ppol induced rDNA damage upon MST2 silencing, accessing UBF translocation to the cap (Fig EV5A). Previous studies and our data in this MS, show that Pol I inhibition in the presence of rDNA DSBs introduced by I-Ppol is fully rescued by ATM inhibition. Under these conditions reduced nucleolar segregation was observed (van Sluis & McStay, 2015 and Harding et al, 2015). Upon MST2 depletion we see a decrease in fully-segregated nucleoli and an increase in partial-segregated nucleoli (Fig EV5A). While this is not as dramatic as ATM inhibition, supports the model of rDNA transcriptional shut down leading to nucleolar segregation. This data also supports a dominant upstream role for ATM that regulates rDNA transcriptional responses via activation of additional downstream factors (Larsen et al., 2014, Ciccia et al., 2014). Of note, in the cases where we see partial-segregated nucleoli we can still observe γ H2Ax, as shown in Figure 7B.

Minor points:

1) Manuscript pages should be numbered, otherwise it's difficult to refer to certain passages of the text

Manuscript pages are now numbered.

2) Somewhere towards the end of the paper, there is this sentence:

"Nucleolar caps are formed under these conditions due to the vast levels of breaks in the nucleoli (van Sluis & McStay, 2015)". This is not entirely correct. Nucleolar caps also form under conditions where few breaks are induced in the rDNA repeats (i.e. upon IR treatment) (see Kruhlak et al., 2007). However, in response to IR, fewer caps form and they are not present in all the nucleoli. Thus, I-Ppo1 treatment just accents this physiological response. This should be clarified in the text.

This now has been changed in the text.

Thank you for submitting your revised manuscript for our consideration. It has now been seen once more by the original reviewers, and I am happy to inform you that all three of them are generally satisfied with the revisions and improvements to the paper. Referee 2 still retains a number of specific issues, of which I would invite you to address the presentational ones (text, microscopy images) as well as the inhibitor control experiments (which could be included in a referee response figure) during a final, minor re-revision round.

 REFEREE REPORTS

Referee #1:

The authors have addressed my concerns. I recommend publication of this manuscript.

Referee #2:

The authors have addressed most of my initial comments and now provide additional experiments, including important controls, to substantiate their findings. In particular the additional data with the H2B S14A mutant and with a second siRNA against MST2 as well as the quantification of the imaging data strengthen the main conclusions. Overall, this is a timely and relevant study, which I consider now well suited for publication in EMBO Journal.

The authors may want to address the following issues to further improve the quality of their work:

- Several microscopy images are still misaligned and the authors may want to correct this prior to publication (e.g. Fig. 2E, 3C, 4F, 5A, 5D). In Fig. 2D in the control condition the green and the red channels do not show the same cells.
- The experiments with DNA-PKi (NU7441), PARPi (olaparib) and Pol Ii (CX-5461) lack the controls that the inhibitors were working.
- A valuable extension of the new RCC1 recruitment data would be to test how this behaves upon MST2 knockdown.
- There are a couple of typos and inconsistencies in the text, which should be corrected, for example:
 - o "HistoneH2B" should be "Histone H2B" in the abstract and throughout the manuscript.
 - o In the abstract, it should be "We show that establishment of H2BS14p ...".
 - o In the introduction, there is a duplication after the first sentence.
 - o Halfway through the first paragraph, "... (Pol I) ATM kinase dependent ..." should probably be rephrased.
 - o Page 4, "phospho-dead" is misspelled
 - o The sentence on the Pol I inhibitor (page 4/5) is unclear. I assume what the authors want to say is that H2BS14p occurs upstream of transcriptional silencing, rather than being induced by Pol I inhibition.
 - o Page 8, fourth paragraph, the sentence on NHEJ is incomplete ("repair" is missing, "in" should be "is").

Referee #3:

I went through the revised version of this manuscript and also carefully read the rebuttal letter. The authors have adequately addressed all of my previous concerns. I therefore recommend publication of this interesting paper without further delay.

We would like to thank all 3 referees for suggesting the publication of our work in EMBO Journal.

Please find below Please find below a point by point replay to the comments raised by Referee 2.

Referee #1:

The authors have addressed my concerns. I recommend publication of this manuscript.

Referee #2:

The authors have addressed most of my initial comments and now provide additional experiments, including important controls, to substantiate their findings. In particular the additional data with the H2B S14A mutant and with a second siRNA against MST2 as well as the quantification of the imaging data strengthen the main conclusions. Overall, this is a timely and relevant study, which I consider now well suited for publication in EMBO Journal.

The authors may want to address the following issues to further improve the quality of their work:

- Several microscopy images are still misaligned and the authors may want to correct this prior to publication (e.g. Fig. 2E, 3C, 4F, 5A, 5D). In Fig. 2D in the control condition the green and the red channels do not show the same cells.

We have now carefully aligned our microscopy images and corrected control condition in Fig 2D.

- The experiments with DNA-PKi (NU7441), PARPi (olaparib) and Pol Ii (CX-5461) lack the controls that the inhibitors were working.

Please find below the data that show that inhibitors are working at the concentrations used in our study. (A) Lack of 5-EU incorporation in CX-4561- treated cells (B) Decreased RPA S4/8 phosphorylation for cells treated with the NU7441 DNA-PK inhibitor (C) Decreased PAR levels in the presence of Olaparib.

- A valuable extension of the new RCC1 recruitment data would be to test how this behaves upon MST2 knockdown.

We agree with the reviewer that an in depth characterisation of how RCC1 recruitment impacts on chromatin condensation in response to DNA damage is an interesting point. We feel however that this is a subject for a separate study and beyond the scope of this manuscript.

- There are a couple of typos and inconsistencies in the text, which should be corrected, for example:

o "HistoneH2B" should be "Histone H2B" in the abstract and throughout the manuscript.

o In the abstract, it should be "We show that establishment of H2BS14p ...".

o In the introduction, there is a duplication after the first sentence.

o Halfway through the first paragraph, "... (Pol I) ATM kinase dependent ..." should probably be rephrased.

o Page 4, "phospho-dead" is misspelled

o The sentence on the Pol I inhibitor (page 4/5) is unclear. I assume what the authors want to say is that H2BS14p occurs upstream of transcriptional silencing, rather than being induced by Pol I inhibition.

o Page 8, fourth paragraph, the sentence on NHEJ is incomplete ("repair" is missing, "in" should be "is").

We have corrected the typos and inconsistencies.

Referee #3:

I went through the revised version of this manuscript and also carefully read the rebuttal letter. The authors have adequately addressed all of my previous concerns. I therefore recommend publication of this interesting paper without further delay.

Corresponding Author Name: Eric O'Neill, Dafni Eleftheria Pefani

Manuscript Number: EMBOJ-2017-98760